# ParalESN: Enabling parallel information processing in Reservoir Computing

## Abstract

Reservoir Computing (RC) has established itself as an efficient paradigm for temporal processing, yet its scalability remains severely constrained by the necessity of processing temporal data sequentially. In this work, we revisit RC through the lens of structured operators and state space modeling, introducing Parallel Echo State Network (ParalESN), a framework that enables the construction of efficient reservoirs with diagonal linear recurrence in the complex space that can be parallelized during training. We provide a theoretical analysis demonstrating that ParalESN preserves the Echo State Property and the universality guarantees of classical Echo State Networks while admitting an equivalent representation of arbitrary linear reservoirs in the complex diagonal form. Empirically, ParalESN attains comparable predictive accuracy to traditional RC on memory and forecasting benchmarks, while delivering substantial gains in training efficiency. On 1-D pixel-level classification tasks, the model achieves competitive accuracy with fully trainable networks, reducing computational costs and energy consumption. Overall, ParalESN offers a promising, scalable, and principled pathway for integrating RC within the deep learning landscape.

## 1 Introduction

Reservoir Computing (RC) has emerged as a simple yet powerful paradigm for harnessing the rich dynamics of recurrent systems for learning and prediction (Nakajima & Fischer, 2021; Lukoševičius & Jaeger, 2009). By fixing a nonlinear recurrent reservoir and training only a linear readout, RC offers favorable training efficiency, strong performance on temporal processing tasks, and intriguing connections to both neuroscience and dynamical systems theory. These properties have led to its success in a wide range of domains, from speech recognition to chaotic signal prediction. Despite these strengths, RC faces the same problem of traditional, fully-trainable Recurrent Neural Networks (RNNs): the input signal has to be processed sequentially, which makes training slow and not parallelizable. In an attempt to improve the efficiency of RC, structured operators have been investigated (Rodan & Tino, 2011; Dong et al., 2020; D'Inverno & Dong, 2025). Additionally, a particularly active research course involve exploring reservoir computing implementations in hardware implementations Gallicchio & Soriano (2025). An important theoretical insight is that even *linear* reservoirs provided that the readout is expressive enough, are universal approximators in the class of fading memory filters (Grigoryeva & Ortega, 2018a;b), thus being able to represent well arbitrary input-output dynamics.

In this work, we address the limitation of classical RC systems by introducing Parallel ESN (ParalESN), a novel class of efficient untrained RNNs with diagonal linear recurrence in the complex space, where the recurrence can easily be parallelized via associative scan. Fig. 1 graphically highlights the proposed model. Our approach rethinks reservoir construction through the lens of structured operators. By combining the dynamical richness of RC with the parallel linear recurrence from Linear Recurrent Units (LRU) (Orvieto et al., 2023), we bridge a gap between classical dynamical systems-inspired learning and contemporary large-scale modeling.

The remainder of this work is organized as follows. Section 2 introduces relevant background on RC and modern sequence models. Section 3 presents the proposed approach, ParalESN. Section 4 presents our theoretical analysis. Section 5 presents the experiments. Finally, Section 6 concludes the paper. Appendix A is dedicated to mathematical proofs. Appendix B provides useful definitions

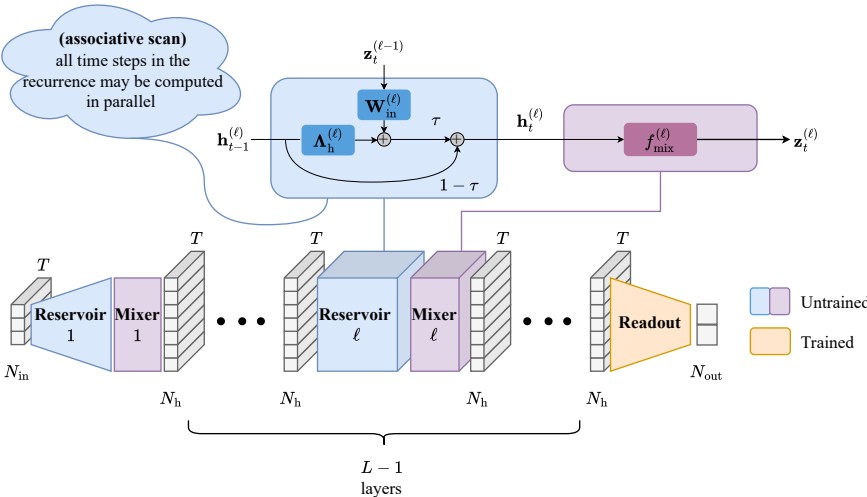

Figure 1: Architectural organization of the proposed ParalESN. The model may have multiple blocks consisting of two components: (i) a linear reservoir and (ii) a non-linear mixing layer. The first block processes the external input as in a traditional shallow architecture. Subsequent blocks process the output of the previous block's mixing layer, $\mathbf{z}_t^{(l-1)}$. The structure of the reservoir (in blue) includes a diagonal, complex-valued recurrent weight matrix $\mathbf{\Lambda}_h^{(\ell)}$ and a dense complex-valued input weight matrix $\mathbf{W}_{in}^{(\ell)}$. The main branch and the temporal residual connections are scaled by a positive coefficient $\tau$ and $1 - \tau$, respectively. The mixing layer (in purple) is used to introduce non-linearity in the model's dynamics and to mix the components of the reservoir states at each time step. The readout (in orange) is the only trainable component in the system. See Section 3 for details.

about concepts used in our theoretical analysis. Appendix C provides details about the benchmarks used in experiments. Appendix D gives details on hyperparameters and model selection, and analyzes the effect on performance of ParalESN's hyperparameters. Appendix E compares ParalESN with other reservoir computing approaches based on structured transforms.

## 2 RELATED WORKS

**Reservoir Computing.** Reservoir Computing (RC) (Verstraeten et al., 2007; Nakajima & Fischer, 2021) is a popular framework for the design of efficient untrained Recurrent Neural Networks (RNNs), developed to address the instabilities of training RNNs (Bengio et al., 1994; Glorot & Bengio, 2010; Pascanu et al., 2013). An RC model consists of two main components: (i) a large recurrent layer, called *reservoir*, that is randomly initialized and then left untrained, and (ii) a trainable readout layer, that may be trained via lightweight closed-form solutions. Therefore, by design, RC models bypass backpropagation and related vanishing/exploding (V/E) gradients, and can be trained exceptionally fast in a single forward pass.

Echo State Networks (ESNs) (Jaeger et al., 2007) established themselves as one of the most successful instance of RC models. The dynamics of an ESN are defined as follows:

$$\mathbf{h}_t = (1 - \tau)\mathbf{h}_{t-1} + \tau\,\sigma(\mathbf{W}_h\mathbf{h}_{t-1} + \mathbf{W}_{in}\mathbf{x}_t + \mathbf{b}). \tag{1}$$

where $\mathbf{h}_t \in \mathbb{R}^{N_h}$ and $\mathbf{x}_t \in \mathbb{R}^{N_{in}}$ are, respectively, the state and the external input at time step $t$. The recurrent weight matrix is denoted as $\mathbf{W}_h \in \mathbb{R}^{N_h \times N_h}$, the input weight matrix is denoted as $\mathbf{W}_{in} \in \mathbb{R}^{N_h \times N_{in}}$, $\mathbf{b} \in \mathbb{R}^{N_h}$ denotes the bias vector, $\sigma$ denotes an element-wise applied nonlinearity, and $\tau \in (0, 1]$ denotes the leaky rate hyperparameter. The entries of the matrix $\mathbf{W}_{in}$, $\mathbf{W}_h$, and $\mathbf{b}$ are generally sampled randomly from a uniform distribution (Gallicchio et al., 2017; Ceni & Gallicchio, 2024) over $(-\omega_{in}, \omega_{in})$ and $(-\omega_b, \omega b)$, respectively. Sampling their entries from Gaussian distributions is also a popular alternative (Verstraeten et al., 2007). The entries of $\mathbf{W}_h$ are

first randomly sampled from a uniform distribution over $(-1, 1)$ and then rescaled to have a desired spectral radius $\rho$[1] In practical applications, the spectral radius is generally constrained to be smaller than 1.

The output of the model is retrieved through the linear readout:

$$\mathbf{y}_t = \mathbf{W}_{\text{out}}\mathbf{h}_t + \mathbf{b}_{\text{out}}. \tag{2}$$

where $\mathbf{y}_t \in \mathbb{R}^{N_{\text{out}}}$ denotes the network output at time step $t$, and $\mathbf{W}_{\text{out}} \in \mathbb{R}^{N_{\text{out}} \times N_h}$ and $\mathbf{b}_{\text{out}} \in \mathbb{R}^{N_{\text{out}}}$ denote the readout weight matrix and bias vector, respectively. The readout is typically optimized via lightweight closed-form solutions, e.g. ridge regression or least squares methods.

The model's state update function defined in equation 1 could depend, in principle, from the initial point $\mathbf{h}_0$. This leads to non-deterministic behaviors, i.e. the impossibility to determine the state solely from the inputs. To avoid that, the reservoir in ESNs is initialized subject to the Echo State Property (ESP) (Yildiz et al., 2012), a useful stability condition for guiding ESNs' initialization. We say that a discrete time dynamical system defined by the transition function $F(\mathbf{h}, \mathbf{x})$ satisfies the ESP if the states asymptotically depends only on the system's inputs. In other words, for any two initial points $\mathbf{h}_0$ and $\mathbf{h}'_0$, then

$$\lim_{t \to \infty} \|F(\mathbf{h}_{t-1} - \mathbf{x}_t) - F(\mathbf{h}'_{t-1}, \mathbf{x}_t)\|_2 = 0. \tag{3}$$

In the literature, it is possible to find ESNs variants that lay their foundation at the intersection of the RC and DL frameworks. In particular, Deep Echo State Networks (DeepESNs) (Gallicchio et al., 2017) represent a class of deep RNN systems where multiple untrained reservoirs are stacked on top of each other. This, together with the architectural bias introduced by increasing the feed-forward depth of the model, has shown promising advantages relative to more traditional, shallow RC approaches. Moreover, notice that DeepESNs generalize the concept of shallow ESNs towards deep architectural constructions. More recently, Residual Echo State Networks (ResESNs) (Ceni & Gallicchio, 2024) introduced a class of RC-based models that leverage residual connections along the temporal dimensions. Its state transition function is defined as follows:

$$\mathbf{h}(t) = \alpha \mathbf{O}\mathbf{h}(t-1) + \beta\phi\Big(\mathbf{W}_h\mathbf{h}(t-1) + \mathbf{W}_x\mathbf{x}(t) + \mathbf{b}\Big), \tag{4}$$

where $\mathbf{O} \in \mathbb{R}^{N_h \times N_h}$ is a random orthogonal matrix, while $\alpha$ and $\beta$ are positive hyperparameters that influence the quality of reservoir dynamics. Interestingly, ResESN can be seen as a generalization of the ESN model, where the linear branch transformation is not constrained to the identity matrix and each branch has independent scaling coefficients (i.e., $\alpha$ and $\beta$).

**Universality of linear reservoirs.** ESP guarantees the universality of the reservoir system in approximating fading memory filters (Grigoryeva & Ortega, 2018a). Provided that the ESP holds, similar universality results have also been proven for reservoirs characterized by linear dynamics, when combined with non-linear readouts that universally approximate functions $f : \mathbb{C}^{N_h} \to \mathbb{C}^{N_y}$ (Grigoryeva & Ortega, 2018b; Gonon & Ortega, 2020). Therefore, linear recurrence ESNs, in combination with non-linear readouts, can approximately arbitrarily well any time-invariant causal filter $U$ with the fading memory property.

**Transformers.** Transformers are the de-facto standard architecture for sequence modeling, managing to replace recurrent models on real-world tasks since their introduction in Vaswani et al. (2017). Differently from RNNs, transformers are a feedforward architecture, and employ self-attention , enabling access to every position of the sequence at any time. This allows for great parallelization and modeling capacity, but comes at a cost of a quadratic complexity in sequence length, making scaling to long contexts challenging. To address this issue, many variants to the naive self-attention aiming to lower the computational and memory burden while maintaining expressivity have been proposed (Tay et al., 2022).

**State Space Models.** Besides training instabilities, another major drawback of stateful sequence models like RNNs is that the input needs to be processed sequentially, greatly limiting the parallelization capabilities of these type of architectures on modern accelerators.

---

[1]The spectral radius of a matrix $\mathbf{A}$, sometimes denoted as $\rho(\mathbf{A})$, is defined as the largest among the lengths of its eigenvalues.

One of the most promising approaches for the parallelization of recurrent models is that of (Deep) State Space Models (SSMs) (Gu et al., 2021). SSMs start from the idea of a linear state space dynamical system, and devise initialization and discretization strategies that help improve memory capacity of the system. Linear recurrence is easily parallelizable (see e.g. Martin & Cundy, 2018), greatly improving training and inference efficiency of this family of models. Structured SSMs such as S4 and S5 (Gu et al., 2021; Smith et al., 2022) demonstrated how linear recurrence, along with a careful initialization based on HiPPO matrices (Gu et al., 2020), can enhance long memory propagation on long sequence tasks. Building on these foundations, Mamba (Gu & Dao, 2024; Dao & Gu, 2024) proposes a selective architecture that allows to change the dynamic based on its inputs. These advances position SSMs as contenders to transformers in sequence modeling, particularly in tasks demanding long memory retention.

**Linear recurrent unit.** Linear recurrent Unit (LRU) (Orvieto et al., 2023) successfully applied linear recurrence to general RNNs, demonstrating that it is possible to deviate from the strict initialization and parametrization rules of SSMs, while retaining their impressive performance. Rather than initializing matrices using HiPPO theory as standard SSMs, LRU employs a diagonal transition matrix, whose eigenvalues are initialized inside the unitary complex disk, using a de-coupled parametrization of their magnitude and phase. In particular, the magnitude is chosen in an interval $[r_{\min}, r_{\max}]$, which allows a finer control over stability and memory capacity of the model. The linear, diagonal structure reduces recurrent updates to parallelizable element-wise operations, and the initialization strategy makes the architecture more stable for longer sequences.

## 3 PARALESN

Inspired by the success of SSMs, we design a more scalable and efficient reservoir design, which we call **ParalESN**. ParalESN improves previous RC models by employing a linear diagonal recurrence, similar to that of LRU, followed by a mixing function that mixes reservoir states. As it is usual in the RC domain, the only trainable part is the readout function. We also explore a *deep* variant of ESNvs, which we label deep ParalESN. See Fig. 1 for a graphical representation of the architecture.

Our approach can be mathematically described as follows:

$$\mathbf{h}_t^{(0)} = \mathbf{x}_t, \tag{5}$$

$$\mathbf{h}_t^{(\ell)} = (1 - \tau)\,\mathbf{h}_{t-1}^{(\ell)} + \tau(\mathbf{\Lambda}_{\mathrm{h}}^{(\ell)}\mathbf{h}_{t-1}^{(\ell)} + \mathbf{W}_{\mathrm{in}}^{(\ell)}\mathbf{z}_t^{(\ell-1)} + \mathbf{b}^{(\ell)}) \text{ for each } \ell \in \{1, \ldots, L\}, \tag{6}$$

$$\mathbf{z}_t^{(\ell)} = f_{\mathrm{mix}}^{(\ell)}(\mathbf{h}_t^{(\ell)}), \tag{7}$$

$$\mathbf{y}_t = f_{\mathrm{readout}}\left(\mathbf{z}_t^{(1)}, \ldots, \mathbf{z}_t^{(L)}\right), \tag{8}$$

where, for each time step $t \in \{1, \ldots, T\}$, $\mathbf{x}_t$ is the external input, $\mathbf{h}_t^{(\ell)}$ is the reservoir state of layer $\ell$, $\mathbf{z}_t^{(\ell)}$ is the hidden state (after mixing), and $\mathbf{y}_t$ is the output. $\mathbf{\Lambda}_{\mathrm{h}}^{(\ell)} \in \mathbb{C}^{N_{\mathrm{h}}}$ is the diagonal, complex-valued recurrent weight matrix, $\mathbf{W}_{\mathrm{in}}^{(\ell)} \in \mathbb{C}^{N_{\mathrm{h}} \times N_{\mathrm{in}}}$ is the complex-valued input weight matrix, $\mathbf{b}^{(\ell)} \in \mathbb{R}^{N_{\mathrm{h}}}$ is the bias vector, and $\mathbf{I} \in \mathbb{R}^{N_{\mathrm{h}} \times N_{\mathrm{h}}}$ is the identity matrix. Similarly to standard ESNs, $\tau$ is the leaky rate. Since the recurrence matrix is diagonal and the recurrence is linear, this term can be added directly to the transition matrix: if we call $\bar{\mathbf{\Lambda}}_h$ the transition matrix without leakage, the transition matrix with leakage can be expressed as $\mathbf{\Lambda}_h = \tau\bar{\mathbf{\Lambda}}_h + (1 - \tau)\mathbf{I}$. The mixer function $f_{\mathrm{mix}}$ is used to introduce non-linearity in the model and to make the components of the hidden state interact. In the mixing layer we consistently use the $\tanh$ non-linearity. In fact, due to the diagonal recurrence, the states all evolve independently before the mixing. We do not mix states at different times, but only the components of the reservoir state at each time steps. For classification of time-series tasks, we only consider the last hidden state of each time series to compute the readout, $\mathbf{y}_T = f_{out}(\mathbf{z}_T)$.

The entries of the input weight matrices and bias vector of the reservoir and mixing layers are, similarly to traditional RC, sampled randomly from a uniform distribution leveraging hyperparameters $\omega_{\mathrm{in}}$, $\omega_{\mathrm{b}}$, $\omega_{\mathrm{mixin}}$, and $\omega_{\mathrm{mixb}}$, respectively. The entries of the diagonal transition matrix are sampled in a way to control its eigenvalues and have a desired minimum and maximum $\rho$ and minimum and maximum angle. Specifically, we sample complex diagonal elements by first drawing radii uniformly from $[\rho_{\min}, \rho_{\max}]$ and phases uniformly from $[\mathrm{phase}_{\min}, \mathrm{phase}_{\max}]$, then converting to complex form and incorporating the leaky rate to shift the eigenvalue center according to $(1 - \tau) + \tau \cdot (\rho e^{i\theta})$.

Fig. 2 graphically demonstrates the speed advantage of the proposed approach by comparing the time required to perform the recurrence between ParalESNs and traditional ESNs. In particular, we observe that the non-parallelizable recurrence of traditional RC is significantly slower compared to that of ParalESN, with execution time scaling linearly with sequence length. In contrast, the parallel implementations exhibit nearly constant execution time regardless of sequence length. Note that, even the deep configuration of ParalESN is consistently faster than the shallow configuration of ESN, despite consisting of an higher number of reservoir layers.

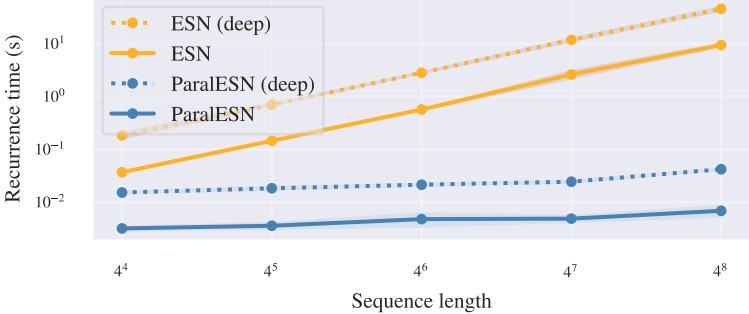

Figure 2: Time required to perform the recurrence by ParalESNs and ESNs for increasing sequence lengths of the input signal, assuming 128 recurrent neurons and 5 layers for deep configurations.

See Appendix E for a comparison between ParalESN and other RC approaches based on structured transforms. ParalESN concistently outperforms other approaches on forecasting benchmarks (Table 8) while achieving lower time complexity, reducing it from linear to logarithmic in the sequence length (Table 9).

## 4 THEORETICAL ANALYSIS

In this section, we will show some theoretical properties of our model. We show a simple condition for the ESP to hold. The ESP is necessary to prove a result of universality for the class of linear reservoir models (Grigoryeva & Ortega, 2018a).

We consider a one-layer ParalESN. Let $\mathbf{W}_{\text{in}} \in \mathbb{C}^{N_h \times N_s}$ be the input-to-hidden matrix, and $\mathbf{\Lambda} \in \mathbb{C}^{N_h \times N_h}$ be the diagonal recurrence matrix. Then, the sequence of hidden states produced by the model given a sequence of inputs $\{\mathbf{x}_1, \ldots, \mathbf{x}_T\} \in (\mathbb{C}^{N_{\text{in}}})^T$ and a starting point $\mathbf{h}_0 \in \mathbb{C}^{N_h}$ is $(\mathbf{y}_0, \ldots, \mathbf{y}_T)$, with

$$\mathbf{h}_t = \begin{cases} \mathbf{h}_0 & \text{if } t = 0 \\ \mathbf{\Lambda}_h \mathbf{h}_{t-1} + \mathbf{W}_{\text{in}} \mathbf{x}_t & \text{otherwise} \end{cases} \tag{9}$$

$$\mathbf{y}_t = f_{\text{out}}(\mathbf{h}_t), \tag{10}$$

where $f_{\text{out}}$ is the readout function. Because the recurrence is linear, the readout function must be non-linear. This definition apparently differs slightly from equations $5 - 8$, but since training does not come into play in this section, we can incorporate $f_{mix}$ and $f_{readout}$ in a single function $f_{out}$.

### 4.1 ECHO STATE PROPERTY

As a first step of the theoretical analysis, we show that it is easy to derive a simple condition for the ESP to hold in the case of linear ESN of equation 9. The same condition on the spectral radius that is necessary for the ESP of ESNs (see, e.g., Jaeger & Haas (2004)) is also sufficient in our case. Moreover, in the case of a diagonal recurrence matrix, we can directly control the spectral radius by looking at the biggest diagonal value in modulus. The following theorem explicitly states this result.

**Theorem 1** (Sufficient and necessary conditions for the ESP). *A ParalESN has the ESP if and only if the diagonal elements $\lambda_1, \ldots, \lambda_{N_h}$ of the recurrence matrix $\mathbf{\Lambda}_h$ are such that for each i, $|\lambda_i| < 1$, where $|\cdot|$ is the complex modulus.*

The proof is given in Appendix A.

## 4.2 EXPRESSIVITY OF PARALESN

An ESN with linear recurrence and MLP readout and the ESP property is universal in the family of fading memory filters, and in particular it is as powerful as non-linear ESNs, that have the same property (Grigoryeva & Ortega, 2018a;b). We have stated the conditions for ParalESN to have the ESP property in Theorem 1. To conclude proving the equivalence of ParalESN with standard ESNs, we now show that an ParalESN is as expressive as a ESN with linear recurrence and any arbitrary transition matrix $\mathbf{W}_h \in \mathbb{C}^{N_h \times N_h}$ and MLP readout.

**Proposition 1.** *Consider an ESN with linear recurrence and a 1-layer MLP readout, defined by*

$$\begin{cases} \mathbf{h}_t = \mathbf{W}_h \mathbf{h}_{t-1} + \mathbf{W}_{in}\mathbf{x}_t \\ \mathbf{y}_t = \mathbf{W}_{out}(\sigma(\mathbf{W}_{hidden}\mathbf{h}_t)) = MLP(\mathbf{h}_t) \end{cases} \tag{11}$$

*where $\mathbf{W}_h$, $\mathbf{W}_{in}$ are matrices of appropriate sizes. Then it exists a ParalESN with MLP readout, defined by equation 9 and equation 10 with $f_{out}$ being another MLP, such that for each input, the two models output the same vector.*

This proposition can be easily proven constructively by diagonalizing the full matrix $\mathbf{A}$. The full proof is given in Appendix A.

By Proposition 1 and results on equivalent expressiveness between ESNs with linear recurrence and standard ESNs, we can deduce that ParalESN and standard ESNs (both with the ESP) are *equivalently expressive*. We explicitly state this fact in the following corollary.

**Corollary 1.** *The class of ParalESN models with the ESP, endowed with an MLP readout, is universal in the family of fading memory filters.*

See Appendix B for definitions related to fading memory filters and the fading memory property.

In practice, in our experiments, the MLP readout is often decomposed into two parts: the first layer (together with the non-linearity) is treated as the fixed mixing function, while only the final linear layer is trained.

## 5 EXPERIMENTS

In this section, we empirically validate the performance of the proposed approach across various benchmarks, including regression and classification tasks on time series. In Section 5.1 we compare the proposed approach with respect to traditional shallow and deep RC on memory-based and forecasting tasks. In Section 5.2 we compare the proposed approach with respect to traditional RC and fully-trainable models on time series and pixel-level 1-D classification tasks. See Appendix C for further details on benchmarks and datasets.

**Methodology.** Model selection is carried out via Bayesian search. See Appendix D for details on the hyperparameters explored for each model and for additional experiments where we explore the effect of ParalESN's hyperparameters on performance. The best configuration is chosen based on the performance achieved on the validation set. For RC models, during model selection, we employ a total number of reservoir neurons of 128 in memory-based and forecasting tasks, and 1024 in classification tasks. Then, we scale up the reservoir as needed for benchmarks on MNIST. The mixing layers are implemented as 1-D convolutions. The readout is either implemented as a ridge regressor trained via Singular Value Decomposition (SVD) solver for memory-based, forecasting, and time series classification tasks, or as a 2-layer Multi-layer Perceptron (MLP) for benchmarks pertaining to the MNIST dataset. Prior to the readout, we always standardize the reservoir's outputs by removing the mean and scaling to unit variance. As our fully-trainable models, we consider a LSTM, a standard Transformer (Vaswani et al., 2017), a Linear Recurrent Unit (LRU) (Orvieto et al., 2023) [2], and Mamba (Gu & Dao, 2024) [3] to cover a wide range of model classes, including recurrent neural networks, attention-based neural networks, and deep state space models, respectively. The LSTM employs a hidden size of 192, the Transformer employs an input size of 96 and a feedforward size of 128 for 3 encoder layers, LRU employs 3 layers with hidden size of 82, Mamba employs 6 layers with model dimension 64, state expansion factor 16, and local convolutional width 4.

---

[2]We use the implementation from github.com/NicolasZucchet/minimal-LRU.
[3]We use the original implementation from github.com/state-spaces/mamba.

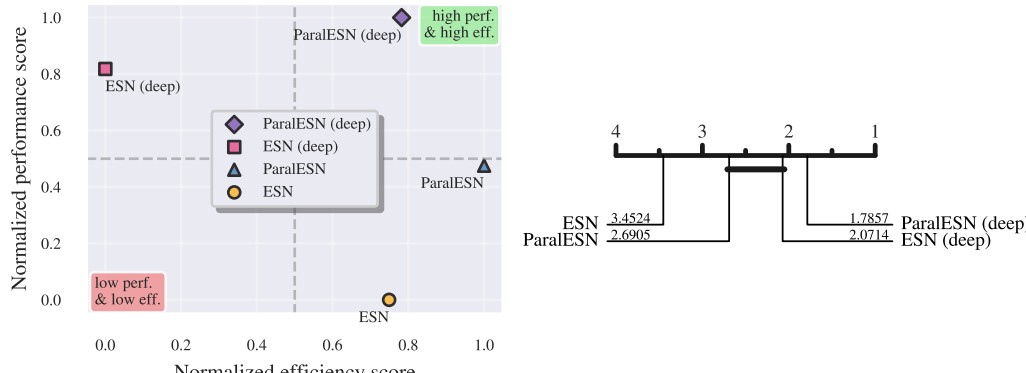

Figure 3: (*left*) Analysis of the trade-off between performance (error for regression tasks and accuracy for classification tasks) and efficiency (training time) for ParalESN and traditional RC, across all considered benchmarks. Performance and efficiency results are the normalized scores, averaged across all tasks. (*right*) Critical difference plot computed via a Wilcoxon test (Demšar, 2006), showing the average rank (lower is better) of ParalESN and traditional RC. Models are ranked based on their overall performance across all benchmarks. Cliques (solid lines) connect models where there is no statistically significant difference in performance.

We train fully-trainable models and the MLP of ParalESN, for at most 100 epochs and with an early-stopping mechanism, to avoid inflating the computational efficiency metrics. For classification, the readout or classification head of each model are fed only the state at the last time step.

**Overall comparison with respect to traditional RC.** Fig. 3 provides an overview of the advantages ParalESN provides with respect to traditional RC, from both performance and efficiency perspectives, presenting the trade-off between performance and efficiency (left) and a critical difference plot ranking models based on their overall performance (right). In the left panel, we observe that ParalESN is both more accurate and more efficient with respect to its counterpart (shallow or deep). In particular, ParalESN (deep), despite consisting of multiple reservoir layers, is able to stay competitive in terms of recurrence speed with respect to a traditional, shallow ESN consisting of just one layer. In the right panel, we note that ParalESN outperforms its shallow counterpart considerably, by a statistically significant margin. Additionally, although no statistically significant difference may be observed between ParalESN (deep) and ESN (deep), the former ranks first as the top-performing model while providing considerable efficiency improvements, as highlighted in the left panel.

### 5.1 Time Series Regression

Memory-based tasks are designed to assess the model's ability to effectively recall delayed versions of the input, while forecasting tasks evaluate the model's ability to predict the time series state at future time steps. Specifically, we consider MemCap (Jaeger, 2001), ctXOR (Verstraeten et al., 2010), and SinMem (Inubushi & Yoshimura, 2017) in our memory-based benchmark. For our forecasting benchmark, we consider Lorenz96 (Lorenz, 1996), Mackey-Glass (MG) (Jaeger & Haas, 2004), NARMA, and a selection of real-world time series forecasting tasks from Zhou et al. (2021), including ETTh1, ETTh2, ETTm1, and ETTm2. See Appendix C.1 and Appendix C.2 for more details on memory-based and forecasting benchmarks, respectively.

**Discussion.** Table 1 and Table 2 present the test set results on memory-based and forecasting tasks, respectively. Fig. 4 graphically compares the training time of ParalESN with respect to traditional RC for each memory-based and forecasting benchmark. Our experiments demonstrate that ParalESN can achieve comparable results to traditional RC across a wide range of time series regression benchmarks, while offering exceptional advantages from a computational efficiency perspective. In all benchmarks, ParalESN trains faster by an entire order of magnitude, except for Lorenz25 and Lorenz50, where the relatively small sequence length reduces the advantage of being able to parallelize the recurrence advantage. Observe that even ParalESN (deep), despite consisting of multiple reservoir layers, trains faster than a traditional, shallow ESN consisting of just one layer. Indeed,

| MEMORY-BASED | MEMCAP ($\uparrow$) | $\cdot 10^{-1}$ CTXOR5 ($\downarrow$) | $\cdot 10^{-1}$ CTXOR10 ($\downarrow$) | $\cdot 10^{-1}$ SINMEM10 ($\downarrow$) | $\cdot 10^{-1}$ SINMEM20 ($\downarrow$) |
|---|---|---|---|---|---|
| ESN | $50.6_{\pm 1.6}$ | $3.6_{\pm 0.1}$ | $7.7_{\pm 0.6}$ | $3.6_{\pm 0.1}$ | $3.7_{\pm 0.1}$ |
| ESN (deep) | $56.8_{\pm 1.3}$ | $\mathbf{3.4}_{\pm \mathbf{0.2}}$ | $5.2_{\pm 1.0}$ | $1.2_{\pm 0.1}$ | $\mathbf{1.6}_{\pm \mathbf{0.1}}$ |
| ParalESN | $115.8_{\pm 1.4}$ | $3.6_{\pm 0.1}$ | $8.2_{\pm 0.1}$ | $3.7_{\pm 0.1}$ | $2.4_{\pm 0.2}$ |
| ParalESN (deep) | $\mathbf{126.0}_{\pm \mathbf{0.1}}$ | $3.6_{\pm 0.1}$ | $\mathbf{5.1}_{\pm \mathbf{0.2}}$ | $\mathbf{1.1}_{\pm \mathbf{0.3}}$ | $2.4_{\pm 0.2}$ |

Table 1: Test set results of memory-based tasks, assuming 128 (total) recurrent neurons for each model. Reported results represent mean and standard deviation over 10 different random initializations. We highlight baselines and our models differently. The **best result** is highlighted in bold.

| FORECASTING | $\cdot 10^{-2}$ Lz25 ($\downarrow$) | $\cdot 10^{-2}$ Lz50 ($\downarrow$) | $\cdot 10^{-4}$ MG ($\downarrow$) | $\cdot 10^{-2}$ MG84 ($\downarrow$) | $\cdot 10^{-2}$ N10 ($\downarrow$) | $\cdot 10^{-2}$ N30 ($\downarrow$) | $\cdot 10^{-1}$ ETTH1 ($\downarrow$) | $\cdot 10^{-1}$ ETTH2 ($\downarrow$) | $\cdot 10^{-1}$ ETTM1 ($\downarrow$) | $\cdot 10^{-1}$ ETTM2 ($\downarrow$) |
|---|---|---|---|---|---|---|---|---|---|---|
| ESN | $10.0_{\pm 0.3}$ | $30.8_{\pm 0.6}$ | $3.0_{\pm 0.0}$ | $6.5_{\pm 0.4}$ | $\mathbf{2.7}_{\pm \mathbf{0.4}}$ | $10.3_{\pm 0.1}$ | $9.1_{\pm 0.2}$ | $13.0_{\pm 1.7}$ | $6.7_{\pm 0.1}$ | $9.9_{\pm 7.3}$ |
| ESN (deep) | $\mathbf{9.7}_{\pm \mathbf{0.2}}$ | $30.5_{\pm 0.3}$ | $\mathbf{2.0}_{\pm \mathbf{0.0}}$ | $\mathbf{4.2}_{\pm \mathbf{0.2}}$ | $3.0_{\pm 0.5}$ | $10.1_{\pm 0.1}$ | $8.9_{\pm 0.1}$ | $9.6_{\pm 0.5}$ | $6.6_{\pm 0.0}$ | $6.0_{\pm 0.6}$ |
| ParalESN | $10.3_{\pm 0.4}$ | $\mathbf{28.8}_{\pm \mathbf{0.4}}$ | $3.0_{\pm 0.0}$ | $7.8_{\pm 0.4}$ | $3.7_{\pm 1.0}$ | $10.3_{\pm 0.1}$ | $8.9_{\pm 0.1}$ | $13.3_{\pm 1.5}$ | $\mathbf{6.5}_{\pm \mathbf{0.0}}$ | $\mathbf{5.0}_{\pm \mathbf{0.1}}$ |
| ParalESN (deep) | $10.4_{\pm 0.4}$ | $\mathbf{28.8}_{\pm \mathbf{0.3}}$ | $3.0_{\pm 0.0}$ | $4.3_{\pm 0.6}$ | $4.4_{\pm 0.7}$ | $10.2_{\pm 0.1}$ | $\mathbf{8.7}_{\pm \mathbf{0.1}}$ | $\mathbf{9.2}_{\pm \mathbf{0.5}}$ | $\mathbf{6.5}_{\pm \mathbf{0.0}}$ | $5.3_{\pm 0.2}$ |

Table 2: Test set results of forecasting tasks, assuming 128 (total) recurrent neurons for each model. Reported results represent mean and standard deviation over 10 different random initializations. We highlight baselines and our models differently. The **best result** is highlighted in bold.

while the readout layer is trained via Ridge regression in both cases, the time required to perform the recurrence through the untrained reservoir is significantly lower in the proposed approach compared to traditional ESNs, thanks to being able to process the sequence in parallel.

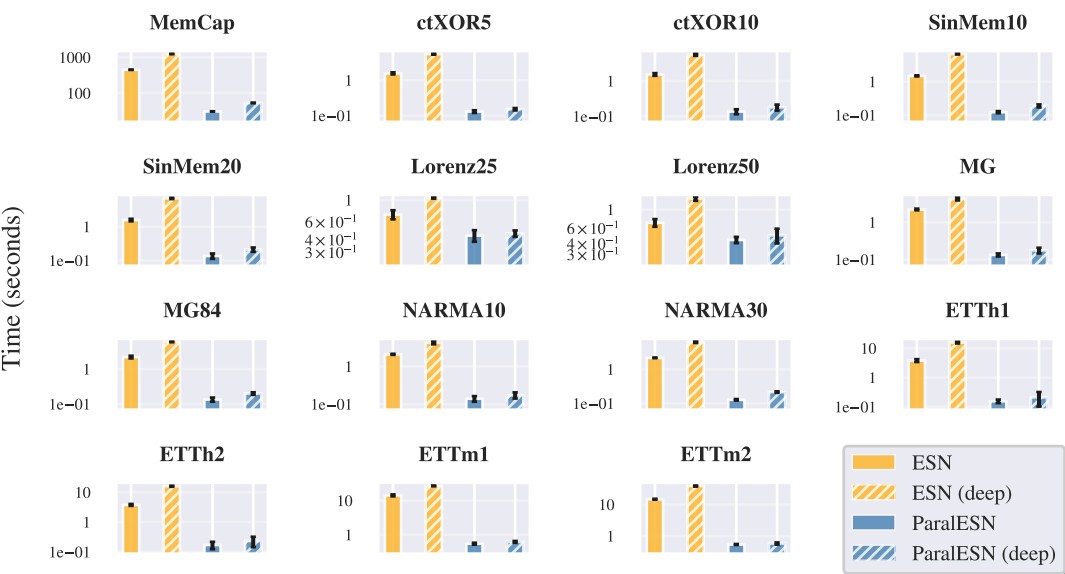

Figure 4: Training time comparison between ParalESNs and traditional RC, for each memory-based and forecasting benchmark. Results are averaged across 10 trials.

## 5.2 TIME SERIES CLASSIFICATION

The time series classification benchmark consists of a selection of tasks from the UEA & UCR repository (Bagnall et al., 2018; Dau et al., 2019). For 1-D pixel-level classification we consider two variations of the MNIST dataset (LeCun, 1998): (i) sequential MNIST (sMNIST), where pixels are flattened into a one-dimensional vector, and (ii) permuted sequential MNIST (psMNIST), where on top of the flattening a random permutation is applied to the pixels. See Appendix C.3 for details.

**Discussion.** Table 3 and Table 4 present test set results on time series classification tasks from the UEA & UCR repository and on sMNIST and psMNIST, respectively. Fig. 5 visualizes the trade-off between performance and efficiency among all models for the MNIST benchmarks. On time series classification tasks, ParalESN demonstrates consistent improvements over traditional ESNs across all benchmarks: +2.63% on FaultDetectionA, +8.79% on FordA, +5.52% on FordB, and +3.58% on StarLightCurvers. We observe performance gains also when comparing their deep configurations, with ParalESN (deep) outperforming ESN (deep) by +7.05% on FaultDetectionA, +1.63% on FordA, +2.90% on FordB, and +3.15% on StarLightCurvers. On sMNIST and psMNIST, ParalESN achieves striking accuracy improvements of +13.41% and +17.13%, respectively, compared to traditional ESN, while ParalESN (deep) provides gains of +5.17% and +15.41% over ESN (deep). These performance improvements come alongside dramatic efficiency gains, with ParalESN and ParalESN (deep) requiring half or less the training time, CO2 emissions, and energy than their counterparts. Additionally, ParalESN stays competitive with fully-trainable models, unlike traditional RC which underperform significantly, without sacrificing its computational advantages.

| MODEL | FAULTDETECTIONA | FORDA | FORDB | STARLIGHTCURVES |
|---|---|---|---|---|
| ESN | $71.93_{\pm 0.85}$ | $75.09_{\pm 1.39}$ | $62.83_{\pm 0.66}$ | $91.82_{\pm 0.65}$ |
| ESN (DEEP) | $84.23_{\pm 0.30}$ | $90.23_{\pm 0.73}$ | $76.20_{\pm 0.95}$ | $93.47_{\pm 0.34}$ |
| PARALESN | $74.56_{\pm 0.69}$ | $83.88_{\pm 2.09}$ | $68.35_{\pm 1.0}$ | $95.40_{\pm 0.44}$ |
| PARALESN (DEEP) | $\mathbf{91.28_{\pm 1.7}}$ | $\mathbf{91.86_{\pm 0.81}}$ | $\mathbf{79.10_{\pm 0.98}}$ | $\mathbf{96.62_{\pm 0.27}}$ |

Table 3: Test set results on time series classification tasks, assuming 1024 recurrent neurons for each model. We highlight baselines and our models differently. The **best result** is highlighted in bold.

**sMNIST**

| MODEL | PARAMS. | ↑ ACCURACY | ↓ TIME (MIN.) | ↓ EMISSIONS (KG) | ↓ ENERGY (KWH) |
|---|---|---|---|---|---|
| LSTM | ≈ 160k | 98.53 | 86.54 | 0.53 | 1.61 |
| TRANSFORMER | ≈ 160k | 98.44 | 154.78 | 1.00 | 3.02 |
| LRU | ≈ 160k | **99.00** | 29.07 | 0.18 | 0.57 |
| MAMBA | ≈ 200k | 98.50 | 29.01 | 0.19 | 0.57 |
| ESN | ≈ 160k | 83.65 | 4.36 | **0.02** | 0.07 |
| ESN (DEEP) | ≈ 160k | 89.67 | 8.30 | 0.05 | 0.14 |
| PARALESN | ≈ 160k | 97.06 | **2.91** | **0.02** | **0.05** |
| PARALESN (DEEP) | ≈ 160k | 94.84 | 3.35 | **0.02** | 0.06 |

**psMNIST**

| MODEL | PARAMS. | ↑ ACCURACY | ↓ TIME (MIN.) | ↓ EMISSIONS (KG) | ↓ ENERGY (KWH) |
|---|---|---|---|---|---|
| LSTM | ≈ 160k | 92.37 | 83.37 | 0.52 | 1.57 |
| TRANSFORMER | ≈ 160k | 97.45 | 155.18 | 1.00 | 3.02 |
| LRU | ≈ 160k | **97.80** | 33.80 | 0.21 | 0.63 |
| MAMBA | ≈ 200k | 92.7 | 37.76 | 0.24 | 0.73 |
| ESN | ≈ 160k | 79.75 | 4.40 | 0.02 | 0.07 |
| ESN (DEEP) | ≈ 160k | 82.24 | 7.41 | 0.04 | 0.12 |
| PARALESN | ≈ 160k | 96.88 | **2.49** | **0.01** | **0.04** |
| PARALESN (DEEP) | ≈ 160k | 97.65 | 2.90 | 0.02 | 0.05 |

Table 4: Test set results sMNIST and psMNIST tasks. We highlight baselines and our models differently. The top-two results are underlined, the **best result** overall is highlighted in bold.

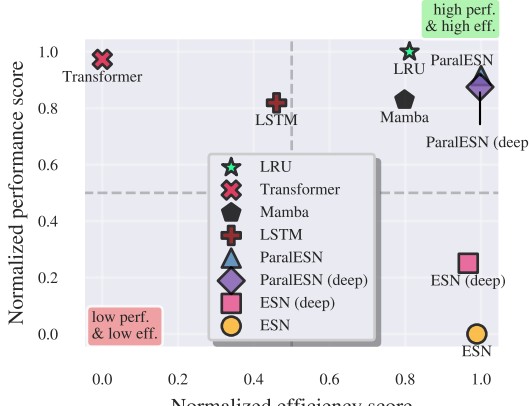

Figure 5: Analysis of the trade-off between performance (test accuracy) and efficiency (training time) for ParalESN, traditional RC, and fully-trainable sequence models, averaged across the sequential MNIST (sMNIST) and permuted sequential MNIST (psMNIST) benchmarks.

## 6 CONCLUSIONS

We introduced ParalESN, a novel framework for the construction of efficient and parallelizable untrained RNNs based on linear diagonal recurrence. This work addresses one of the fundamental limitations of traditional RC, where the input must be processed sequentially, resulting in slow training. Results across various benchmarks on time series and 1-D pixel-level classification showcase that ParalESN is, on average, more accurate and faster than traditional RC, while staying competitive with fully-trainable sequence models from deep learning.

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

## A PROOFS

### A.1 PROOF OF THEOREM 1

*Proof.* Let us start proving the sufficient condition. Let $\mathbf{h}_0$ $\mathbf{z}_0$ be two vectors in $\mathbb{C}^{N_h}$, and $\mathbf{x}_1, \ldots, \mathbf{x}_N$ be a sequence of inputs. Let $\rho(\mathbf{\Lambda}_h) < 1 - \epsilon$ be the spectral radius of $\mathbf{\Lambda}_h$. Because $\mathbf{\Lambda}_h$ is diagonal, it holds $\rho(\mathbf{\Lambda}_h) = \|\mathbf{\Lambda}_h\|_2$. Then,

$$
\begin{aligned}
\|\mathbf{h}_N - \mathbf{z}_N\|_2 &= \|\mathbf{\Lambda}_h \mathbf{h}_{N-1} + \mathbf{W}_{in}\mathbf{x}_N - (\mathbf{\Lambda}_h \mathbf{z}_{N-1} + \mathbf{W}_{in}\mathbf{x}_N)\|_2 \\
&= \|\mathbf{\Lambda}_h(\mathbf{h}_{N-1} - \mathbf{z}_{N-1})\|_2 \\
&\leq \|\mathbf{\Lambda}_h\|_2 \cdot \|(\mathbf{h}_{N-1} - \mathbf{z}_{N-1})\|_2 \\
&= \rho(\mathbf{\Lambda}_h) \cdot \|(\mathbf{h}_{N-1} - \mathbf{z}_{N-1})\|_2 \\
&\leq (1 - \epsilon) \cdot \|(\mathbf{h}_{N-1} - \mathbf{z}_{N-1})\|_2 \\
&\vdots \\
&\leq (1 - \epsilon)^N \cdot \|\mathbf{h}_0 - \mathbf{z}_0\| \xrightarrow{N \to \infty} 0.
\end{aligned}
$$

This proves the sufficient condition. To prove the necessary condition, suppose that some diagonal entry $\lambda_i$ of of $\mathbf{\Lambda}_h$ is greater than 1. Then given two initial states $\mathbf{h}_0, \mathbf{z}_0$ such that $(\mathbf{h}_0)_i \neq (\mathbf{z}_0)_i$, unrolling the linear recursion we get the explicit formulas for $\mathbf{h}_N$ and $\mathbf{z}_N$:

$$
\mathbf{h}_N = \mathbf{\Lambda}_h^N \mathbf{h}_0 + \sum_{i=1}^{N-1} \mathbf{\Lambda}_h^{N-i} \mathbf{x}_i \tag{12}
$$

$$
\mathbf{z}_N = \mathbf{\Lambda}_h^N \mathbf{z}_0 + \sum_{i=1}^{N-1} \mathbf{\Lambda}_h^{N-i} \mathbf{x}_i \tag{13}
$$

Subtracting the two terms, we obtain

$$
\mathbf{h}_N - \mathbf{z}_N = \mathbf{\Lambda}_h^N (\mathbf{h}_0 - \mathbf{z}_0). \tag{14}
$$

Focusing on component $i$, we obtain that $(\mathbf{h}_N)_i - (\mathbf{z}_N)_i = \lambda_i^N((\mathbf{h}_0)_i - (\mathbf{z}_0)_i)$. Since $(\mathbf{z}_0)_i \neq (\mathbf{h}_0)_i$ and $\lambda_i > 1$, this term never reaches 0, growing exponentially in magnitude.

$\square$

### A.2 PROOF OF PROPOSITION 1

*Proof.* Every square matrix is diagonalizable in the complex field, up to arbitrarily small perturbations in its entries, so we can write $\mathbf{W}_h = \mathbf{V}\mathbf{\Lambda}_h \mathbf{V}^{-1}$, with $\mathbf{V} \in \mathbb{C}^{N_h \times N_h}$, and $\mathbf{\Lambda}_h$ a diagonal matrix, having on the diagonal the eigenvalues of $\mathbf{W}_{in}$. We can rewrite the recurrence in terms of $\mathbf{V}$ and $\mathbf{\Lambda}$ as

$$
\mathbf{h}_t = \mathbf{V}\mathbf{\Lambda}_h \mathbf{V}^{-1}\mathbf{h}_{t-1} + \mathbf{W}_{in}\mathbf{x}_t. \tag{15}
$$

Pre-multiplying each side of equation 15 by $\mathbf{V}^{-1}$, we get a complex diagonal ESN having the equations

$$
\begin{cases}
\tilde{\mathbf{h}}_t = \mathbf{\Lambda}_h \tilde{\mathbf{h}}_{t-1} + \tilde{\mathbf{W}}_{in}\mathbf{x}_t \\
\mathbf{y}_t = \mathbf{W}_{out}(\sigma(\tilde{\mathbf{W}}_{hidden}\tilde{\mathbf{h}}_t))
\end{cases}
$$

where $\tilde{\mathbf{h}}_t = \mathbf{V}^{-1}\mathbf{h}_t$, $\tilde{\mathbf{W}}_{in} = \mathbf{V}^{-1}\mathbf{W}_{in}$, and $\tilde{\mathbf{W}}_{hidden} = \mathbf{W}_{hidden}\mathbf{V}$.

Therefore, for any possible linear ESN and for any sequence of inputs, it exists an equivalent complex diagonal ESN that, when given the same inputs, produces the same outputs.

$\square$

## B FILTERS AND FADING MEMORY PROPERTY

In this appendix, we provide a broad overview of the definitions used to characterize the class of filters that ESNs and linear ESNs can universally approximate. Specifically, we introduce the concept

of fading memory filters We will mainly follow (Grigoryeva & Ortega, 2018a) and we point to it for further details.

**Filters.** Informally, a filter is a function that takes (semi)infinite sequences and returns (semi)infinite sequences. More precisely, given $N_x \in \mathbb{N}$ and $N_y \in \mathbb{N}$, we define a filter $\mathcal{U}$ as follows:

$$\mathcal{U} : (\mathbb{R}^{N_x})^{\mathbb{Z}_-} \to (\mathbb{R}^{N_x})^{\mathbb{Z}_-}, \tag{16}$$

where $(\mathbb{R}^N)^{\mathbb{Z}_-}$ is the set containing the semi-infinite sequences of vectors of dimension $N$, that are indexed by negative integer numbers (0 included)[4], $\overleftarrow{\mathbf{v}} = (\mathbf{v}_i)_{i \in \mathbb{Z}_-}$ with $\mathbf{v}_i \in \mathbb{R}^N$. Two useful properties we would like for filters are *causality* and *time-invariance*. A filter $\mathcal{U}$ is causal if its input at time $t$, $\mathbf{y}_t = \mathcal{U}(\overleftarrow{\mathbf{x}})_t$ only depends on the inputs up until time $t$, i.e. $\overleftarrow{\mathbf{x}}_{(-\infty, t]}$. Causality of the target function is an intuitively useful property in tasks such as forecasting or autoregressive generation: without this property, seeing the past inputs could not be sufficient to determine univocally the present output.

A filter is said to be time-invariant if, when given in input two shifted inputs, it outputs two sequences shifted by the same amount. Formally, for any positive integer $\tau > 0$, we can define the *time-shift operator* $\mathcal{T}_\tau$ as $\mathcal{T}_\tau(\overleftarrow{\mathbf{x}})_t = \overleftarrow{\mathbf{x}}_{t-\tau}$. A filter is time-invariant if it commutes with the time-shift operator. This property is also useful, because it makes so that the filter definition does not depend explicitly on the time $t$.

**Infinite norm.** The space $(\mathbb{R}^N)^{\mathbb{Z}}$ can be endowed with a norm, giving us a notion of distance. In particular, given a the usual euclidean norm for vectors $\|\cdot\|$, we define the *infinite norm* of a sequence as the supremum of the norm of any vector of the sequence.

$$\|\overleftarrow{\mathbf{v}}\|_\infty = \sup_{i \in \mathbb{Z}} \|\mathbf{v}_i\|. \tag{17}$$

A *distance* between two sequences $\overleftarrow{\mathbf{v}}$ and $\overleftarrow{\mathbf{w}}$ can then be defined as the infinite norm of their difference:

$$d_\infty(\overleftarrow{\mathbf{v}}, \overleftarrow{\mathbf{w}}) = \|\overleftarrow{\mathbf{v}} - \overleftarrow{\mathbf{w}}\|_\infty. \tag{18}$$

**Weighted norm:** Intuitively, when measuring the difference between two sequences, often we would like to "count" more recent values more than ones distant in the past. Since infinite norm counts all values equally, this norm cannot achieve this important requirement. Therefore, we introduce a *weighting sequence* $w = (w)_i \in (0, 1]^{\mathbb{Z}_-}$, such that $\lim_{i \to -\infty} w_i = 0$, and define the *weighted norm* as:

$$\|\overleftarrow{\mathbf{v}}\|_w = \|w \odot \overleftarrow{\mathbf{v}}\|_\infty, \tag{19}$$

where $\odot$ is the element-wise product.

**Fading memory property:** Finally, we can give a formal definition of fading memory property. The class of filters with fading memory property is the family of function that ESNs, and their linear counterparts, approximate arbitrarily well, thus making them universal. A fading memory filter is, simply put, a filter that is continuous when considering the distance induced by any weighted norm $\|\cdot\|_w$:

**Definition 1.** A filter $\mathcal{U} : (\mathbb{R}^{N_x})^{\mathbb{Z}_-} \to (\mathbb{R}^{N_y})^{\mathbb{Z}_-}$ has the fading memory property if for any $\overleftarrow{\mathbf{x}}_1 \in (\mathbb{R}^{N_x})^{\mathbb{Z}_-}$ and $\epsilon > 0$, it exists $\delta = \delta(\epsilon) > 0$ such that if for any $\overleftarrow{\mathbf{x}}_2 \in (\mathbb{R}^{N_x})^{\mathbb{Z}_-}$ that satisfies

$$\|\overleftarrow{\mathbf{x}}_1 - \overleftarrow{\mathbf{x}}_2\|_w < \delta,$$

then

$$\|\mathcal{U}(\overleftarrow{\mathbf{x}}_1) - \mathcal{U}(\overleftarrow{\mathbf{x}}_2)\|_w < \epsilon \tag{20}$$

ESNs have been proved universal in approximating arbitrarily well *time-invariant, causal filters with the fading memory property* (Grigoryeva & Ortega, 2018a) (Theorem 4.1).

---

[4]We could also define filters for infinite sequence, i.e. with inputs and outputs indexed by $\mathbb{Z}$. However, since in practice we are mainly interested to the "last" point of the sequences, it is common to restrict the definition to semi-infinite sequences.

# C DATASETS

Here, we provide more context and details for each benchmark considered in our experiments.

## C.1 MEMORY-BASED

**MemCap.** The MemCap task involves outputting a delayed version of the input $k$ time steps in the past. The memory capacity score, used to assess performance on the MemCap task, is computed by summing the squared correlation coefficients between the $k$-th time step delayed version of the input and the output for each delay value $k = 1, \ldots, 200$. We generate an input sequence uniform in $[-0.8, 0.8]$ with length $T = 7000$. The first 5000 time steps are used for training, the next 1000 for validation, and the final 1000 for testing. Both training and inference phases employ a 100 time step washout to warm up the reservoir.

**ctXOR.** Consider a one-dimensional input time series $\mathbf{x}(t)$ uniformly distributed in $(-0.8, 0.8)$ and assume $\mathbf{r}(t - d) = \mathbf{x}(t - d - 1)\mathbf{x}(t - d)$. The task is to output the time series $\mathbf{y}(t) = \mathbf{r}(t - d)^2 \operatorname{sign}(\mathbf{r}(t - d))$, where $d$ is the delay and $p$ determines the strength of the non-linearity. We consider delay $d = 5$ (ctXOR5) and delay $d = 10$ (ctXOR10).

**SinMem.** Given a one-dimensional input time series $\mathbf{x}(t)$ uniformly distributed in $(-0.8, 0.8)$, the task is to output the time series $\mathbf{y}(t) = \sin(\pi \mathbf{x}(t - d))$. For SinMem, we consider delays $d = 10$ (SinMem10) and $d = 20$ (SinMem20).

## C.2 FORECASTING

**Lorenz96.** The Lorenz96 (Lz) task is to predict the next state of the time series $\mathbf{x}(t)$, expressed as the following 5-dimensional chaotic system:

$$\mathbf{x}(t) = \frac{\partial}{\partial t} f_i(t) = f_{i-1}(t)(f_{i+1}(t) - f_{i-2}(t)) - f_i(t) + 8, \tag{21}$$

for $i = 1, \ldots, 5$. In our experiments, we focus on predicting the 25-th (Lz25) and 50-th (Lz50) future state of the time series. Thus, the task involves predicting $\mathbf{y}(t) = \mathbf{x}(t + 25)$ and $\mathbf{y}(t) = \mathbf{x}(t + 50)$, respectively. We generate a time series of length $T = 1200$. The first 400 time steps are used for training, the next 400 for validation, and the final 400 for testing.

**Mackey-Glass.** The Mackey-Glass 17 (MG) task is to predict the next state of the following time series:

$$\mathbf{x}(t) = \frac{\partial}{\partial t} f(t) = \frac{0.2 f(t - 17)}{1 + f(t - 17)^{10} - 0.1 f(t)}. \tag{22}$$

In our experiments, we focus on predicting the 1-st and 84-th future state of the time series. Thus, the task involves predicting $\mathbf{y}(t) = \mathbf{x}(t + 1)$ (MG) and $\mathbf{y}(t) = \mathbf{x}(t + 84)$ (MG84), respectively. We generate a time series of length $T = 10000$, with the first 5000 time steps used for training, the next 2500 for validation, and the final 2500 for testing.

**NARMA.** Given a one-dimensional input time series $\mathbf{x}(t)$ uniformly distributed in $[0, 0.5]$, the NARMA task is to predict the next state of the following time series:

$$\mathbf{y}(t) = 0.3\mathbf{y}(t - 1) + 0.01\mathbf{y}(t - 1) \sum_{i=1}^{t=d} \mathbf{y}(t - i) + 1.5\mathbf{x}(t - d)\mathbf{x}(t - 1) + 0.1. \tag{23}$$

We will consider the NARMA30 (N30) and NARMA60 (N60), with (look-ahead) delay of $d = 30$ and $d = 60$, respectively. We generate a time series of length $T = 10000$, with the first 5000 time steps used for training, the next 2500 for validation, and the final 2500 for testing.

**Real-world datasets.** We consider four publicly available datasets from (Zhou et al., 2021), ETTh1, ETTh2, ETTm1, and ETTm2. In Table 5 an overview of their characteristics. These represent real-world, challenging forecasting tasks. We focus on the multivariate case and predict all provided features. We use a delay (lookback window) of 192 timesteps. We normalize each feature in the training set independently to zero mean and unit variance. The normalization coefficients computed on the training data are then applied to normalize the validation and test data. Since we observe order-of-magnitude outliers in the training set, we clip the normalized training data to the interval $(-10, 10)$ across all datasets. The validation and test data remain unmodified.

| DATASET | TRAIN | VALIDATION | TEST | NO. OF FEATURES |
|---------|-------|------------|------|-----------------|
| ETTH1/2 | 8640 | 2880 | 2880 | 7 |
| ETTM1/2 | 34560 | 11520 | 11520 | 7 |

Table 5: Overview of the real-world forecasting datasets tasks. Training, validation, and test sizes are measured in the number of time steps.

## C.3 CLASSIFICATION

In Table 6 an overview of considered classification datasets. The validation set is obtained via a $70 - 30$ stratified split for time series classification tasks, and via a $90 - 10$ stratified split for sMNIST and psMNIST. All datasets are one-dimensional (i.e., only one feature per time step). No data augmentation is applied.

| DATASET | TRAIN | TEST | LENGTH | NO. OF CLASSES |
|---------|-------|------|--------|----------------|
| FAULTDETECTIONA | 10912 | 2728 | 5120 | 3 |
| FORDA | 3601 | 1320 | 500 | 2 |
| FORDB | 3636 | 810 | 500 | 2 |
| SMNIST/PSMNIST | 60000 | 10000 | 784 | 10 |
| STARLIGHTCURVERS | 1000 | 8236 | 1024 | 3 |

Table 6: Overview of time series and 1-D pixel-level classification tasks.

# D HYPERPARAMETERS & MODEL SELECTION

## D.1 MODEL SELECTION

Here, we provide details on the hyperparameters explored for each model class.

For RC, common hyperparameters explored for both ESN and ParalESN include the input scaling $\omega_{\text{in}}$, the bias scaling $\omega_{\text{b}}$, and the leaky rate $\tau$. For traditional RC we explore the spectral radius $\rho$, used to rescale the recurrent weight matrix, while for ParalESN we explore the minimum and maximum magnitude of the diagonal entries $\rho_{\text{min}}$, $\rho_{\text{max}}$, and the minimum and maximum phase. i.e. the angle with the positive x-axis, $\text{phase}_{\text{min}}$, and $\text{phase}_{\text{max}}$, used to sample the diagonal transition matrix and easily control its eigenvalues. Additionally, we explore $\omega_{\text{mixin}}$ and $\omega_{\text{mixb}}$ for scaling the input weight matrix and the bias vector of the mixing layer, and $k$ for determining the kernel size. For the deep configurations, we explore the number of (untrained) reservoir layers $N_{\text{L}}$ and we explore an additional set of hyperparameters for layers beyond the first one, to promote diverse dynamics between the first layer and deeper ones. Additionally, we consider hyperparameter `concat` $\in [\text{False}, \text{True}]$, which determines whether the readout is fed the states of the last layer or the concatenation of the states across all layers. To ensure a consistent number of trainable parameters when states are concatenated, the hidden size (i.e., the total number of reservoir units) is evenly split across layers. If an even split is not possible, the remaining neurons are allocated to the first layer. When the readout is implemented as a ridge regressor, we explore regularization strength values in $[0, 0.01, 0.1, 1, 10, 100]$. When the readout is implemented as an MLP, we simply train it with the Adam optimizer with a learning rate of $5e - 4$; the rest are the default PyTorch values. See Table 7 for an overview of the values explored.

For fully-trainable models, we employ the Adam optimizer and sweep through the learning rate with a log uniform distribution over the interval $[0.0001, 0.01]$, and the weight decay exploring values in $[0, 1e - 7, 1e - 6, 1e - 5, 1e - 4, 1e - 3, 1e - 2, 1e - 1]$. For LRU, we also sweep through $\rho_{\text{min}}$ with uniform distribution over the interval $[0, 1]$, $\rho_{\text{max}}$ with uniform distribution over the interval $[0.8, 1]$, and $\text{phase}_{\text{max}}$ with uniform distribution over the interval $[0.001, 3.14]$.

| HYPERPARAMETERS | VALUES |
|---|---|
| concat | [False, True] |
| $N_L$ | [1, 2, 3, 4, 5] |
| $\omega_{in}$ and inter-$\omega_x$ | [0.01, 0.1, 1, 10] |
| $\omega_b$ and inter-$\omega_b$ | [0, 0.01, 0.1, 1, 10] |
| $\tau$ and inter-$\tau$ | [0.1, 0.5, 0.9, 1] |
| **(ESN)** | |
| $\rho$ and inter-$\rho$ | [0.1, 0.5, 0.9] |
| **(ParalESN)** | |
| *(Reservoir layer)* | |
| $\rho_{max}$ and inter-$\rho_{max}$ | [0.1, 0.5, 0.9] |
| $\rho_{min}$ and inter-$\rho_{min}$ | [0.1, 0.5, 0.9] * |
| phase$_{max}$ and inter-phase$_{max}$ | [$\frac{1}{2}\pi$, $\pi$, $2\pi$] |
| phase$_{min}$ and inter-phase$_{min}$ | [0, $\frac{1}{2}\pi$, $\pi$, $2\pi$] * |
| *(Mixing layer)* | |
| $\omega_{mixin}$ and inter-$\omega_{mixin}$ | [0.01, 0.1, 1, 10] |
| $\omega_{mixb}$ and inter-$\omega_{mixb}$ | [0, 0.01, 0.1, 1, 10] |
| $k$ and inter-$k$ | [3, 5, 7, 9] |

   * Min value must be smaller or equal than the max.

Table 7: Model selection hyperparameters for RC models.

## D.2 HYPERPARAMETERS' EFFECT

In Fig. 6 we explore the effect on performance of ParalESN's main hyperparameters, including spectral radius, phase, leaky rate, and initialization strategy for the input weight matrix.

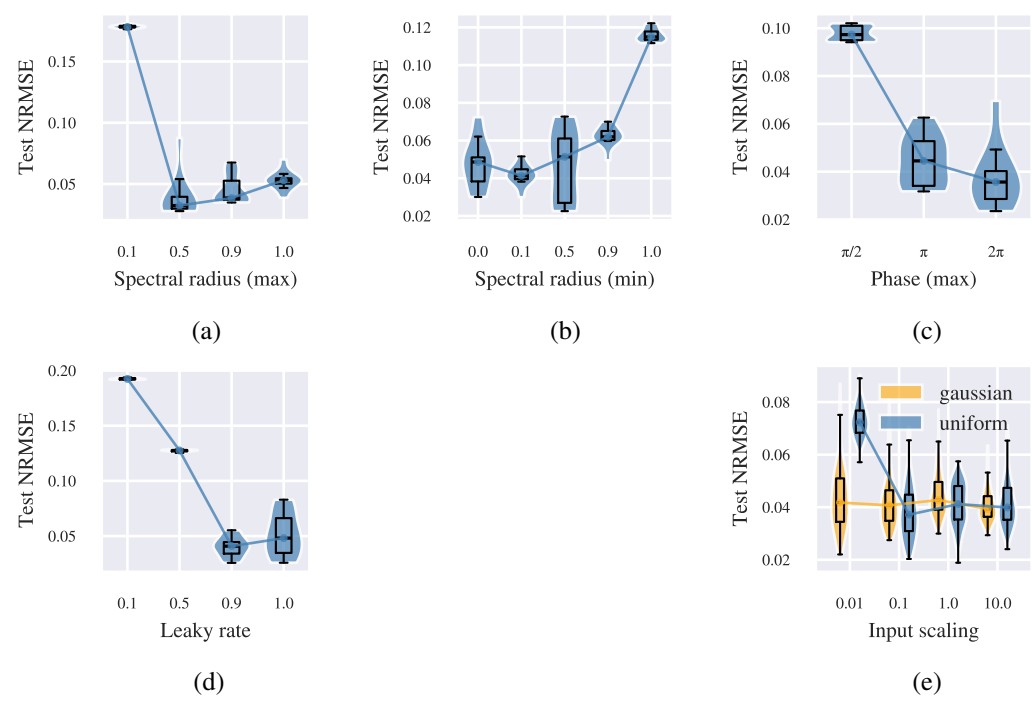

Figure 6: Effect of ParalESN's hyperparameters on the NARMA10 benchmark: (a-b) spectral radius, (c) phase, (d) leaky rate, and (e) input scaling and initialization strategy. For the Gaussian init., values are drawn from $\mathcal{N}(0, \text{scaling}^2)$. Violin represents evaluation on 10 runs.

| FORECASTING | $\cdot 10^{-2}$ Lz25 ($\downarrow$) | $\cdot 10^{-2}$ Lz50 ($\downarrow$) | $\cdot 10^{-4}$ MG ($\downarrow$) | $\cdot 10^{-2}$ MG84 ($\downarrow$) | $\cdot 10^{-2}$ N10 ($\downarrow$) | $\cdot 10^{-2}$ N30 ($\downarrow$) |
|---|---|---|---|---|---|---|
| ESN | $\mathbf{10.0}_{\pm 0.3}$ | $30.8_{\pm 0.6}$ | $\mathbf{3.0}_{\pm 0.0}$ | $\mathbf{6.5}_{\pm 0.4}$ | $\mathbf{2.7}_{\pm 0.4}$ | $\mathbf{10.3}_{\pm 0.1}$ |
| SCR | $22.1_{\pm 0.4}$ | $35.6_{\pm 0.8}$ | $18.7_{\pm 4.2}$ | $32.2_{\pm 3.1}$ | $11.1_{\pm 0.2}$ | $16.1_{\pm 0.9}$ |
| Structured RC | $10.4_{\pm 0.2}$ | $30.9_{\pm 0.5}$ | $12.6_{\pm 0.1}$ | $28.5_{\pm 1.6}$ | $18.7_{\pm 0.1}$ | $20.9_{\pm 0.1}$ |
| ParalESN | $10.3_{\pm 0.4}$ | $\mathbf{28.8}_{\pm 0.4}$ | $\mathbf{3.0}_{\pm 0.0}$ | $7.8_{\pm 0.4}$ | $3.7_{\pm 1.0}$ | $\mathbf{10.3}_{\pm 0.1}$ |

Table 8: Comparison of structured reservoir computing approaches on time series forecasting, assuming 128 recurrent neurons for each model. A traditional ESN is included as a baseline. Reported results represent mean and standard deviation over 10 different random initializations. We highlight baselines and our models differently. Top-two results are underlined, **best result** is in bold.

## E    COMPARISON WITH STRUCTURED RESERVOIR COMPUTING

Here, we compare ParalESN with other reservoir computing approaches based on structured transforms on time series forecasting benchmarks and in terms of their time complexity. We consider Simple Cycle Reservoir (SCR) (Rodan & Tino, 2011) and Structured Reservoir Computing (Structured RC) (Dong et al., 2020)[5]. The former replaces the full transition matrix $\mathbf{W}_h$ in equation 1 with a fixed ring topology. The latter replaces the transition matrix with a composition of Hadamard and diagonal matrices. Despite not being based on structured transforms, we report traditional ESN results as a baseline.

### E.1    TIME SERIES FORECASTING

Table 8 presents the test set results of on the forecasting benchmarks, for ParalESN, SCR, Structured RC, and ESN. For simplicity, we consider a shallow configuration of 1 layer for all models. ParalESN is consistently one of the top-performing models along with ESN, while other approaches based on structured transforms such as SCR and Structured RC consistently underperform across all datasets. In particular, compared to SCR and Structured RC, ParalESN achieves lower test error on MG, MG84, and N10 by an entire order of magnitude and half the error on N30.

Model selection for both SCR and Structured RC is performed using the same methodology employed for the other models, as described in Appendix D.1. However, rather than employing Bayesian search, we use grid search due to the relatively contained number of configurations. We explore the same hyperparameters and intervals highlighted in Table 7 for traditional ESNs. Note that in SCR, there is no need to tune the spectral radius since the recurrent weight matrix is fixed. In Structured RC, rather than tuning the spectral radius of the recurrent weight matrix, we tune the reservoir scaling for values in $[0.01, 0.1, 1, 10]$.

### E.2    COMPUTATIONAL COMPLEXITY

Table 9 provides an overview of the time complexity of computing the recurrence in ParalESN, SCR, Structured RC, and traditional ESNs. We include the complexity for performing a single step of recurrence and processing the entire sequence. In comparison to standard ESNs, and to other structured approaches, the diagonal transition matrix of ParalESN allows for a number of operation that scales linearly in the reservoir size $N_h$. Furthermore, the linear recurrence, which makes it possible to compute all time steps in parallel via associative scan, avoids the linear dependency on the sequence length which becomes logarithmic.

**ESN.** Let $\{x_1, \ldots, x_L\}$ be an input sequence of length $L$, where each $x_i \in \mathbb{R}^{N_i}$. A standard nonlinear ESN with $N_h$ hidden units updates its reservoir state according to Equation 1. Each time step requires two matrix–vector multiplications: one with the recurrent weight matrix ($N_h \times N_h$) and one with the input weight matrix ($N_h \times N_i$). Thus, the total computational cost over the entire sequence is:

$$\mathcal{O}\big(L(N_h^2 + N_h N_i)\big) = \mathcal{O}(LN_h^2),$$

where the final equality assumes $N_i < N_h$, as it is usual in ESNs.

---

[5]We use the original implementation from github.com/rubenohana/Reservoir-computing-kernels.

| MODEL | SINGLE STEP | WHOLE SEQUENCE |
|---|---|---|
| ESN | $\mathcal{O}(N_h^2)$ | $\mathcal{O}(LN_h^2)$ |
| SCR | $\mathcal{O}(N_h N_i)$ | $\mathcal{O}(LN_h N_i)$ |
| STRUCTURED RC | $\mathcal{O}(N_h \log N_h)$ | $\mathcal{O}(LN_h \log N_h)$ |
| PARALESN | $\mathcal{O}(N_h N_i)$ | $\mathcal{O}(\log(L)N_h N_i)$ |

Table 9: Overview of the time complexity analysis. We highlight baselines and our models differently.

**Simple Cycle Reservoir.** Another notable example of RC model that uses structured transformations is the Simple Cycle Reservoir (SCR), in which the hidden units are arranged in a ring topology. The state update can be implemented in $\mathcal{O}(N_h)$ time (including the multiplication of the state for a scaling factor $\rho$), leading to the same complexity as a sequential ParalESN:

$$\mathcal{O}(LN_h N_i).$$

**Structured reservoir computing.** Structured reservoir computing replaces the full transition matrix $\mathbf{W}_h$ in equation 1 with structured transformations to reduce the computational cost. Specifically, a composition of Hadamard and diagonal matrices, reducing the time complexity by a $\log$ factor in the reservoir size and achieving:

$$\mathcal{O}(LN_h \log N_h),$$

assuming $N_i < \log N_h$.

In the same work, Dong et al. (2020) also introduce a *recurrent kernel* approach, that trades off the dependency on the reservoir size for a superlinear dependency on the sequence length, given by the necessity of computing a $L \times L$ kernel matrix. This approach goes in the opposite direction of ours, being more suitable for shorter sequences, where the number of reservoir units may be larger than the sequence length.

**ParalESN.** In our approach, the reservoir update uses a diagonal transition matrix, which reduces the per-step computation to:

$$\mathcal{O}(N_h N_i).$$

Hence, the sequential computation over a length-$L$ sequence has total cost:

$$\mathcal{O}(LN_h N_i).$$

Furthermore, because state updates are linear, the computation can be parallelized along the temporal dimension. Using a parallel associative scan algorithm (Martin & Cundy, 2018), the dependence on $L$ is reduced to a logarithmic factor[6], yielding an overall complexity of:

$$\mathcal{O}\left(\log(L) N_h N_i\right).$$

Note that, the associative scan could in principle be applied to any linear recurrence RC model, but practical limitations remain. Because the algorithm involves repeatedly squaring the transition matrix, it would in general require $N_h^3$ operations for scan step, for full, unstructured matrices. Moreover, in the case of Structured RC, the matrix-matrix multiplication would create a full matrix, destroying the Hadamard structure.

---

[6]Assuming $\Theta(L/\log L)$ parallel processors.

