# OpenReview forum: "ESNv2: Resurrecting Reservoir Computing in the Deep Learning era"
_ICLR.cc/2026/Conference — Submitted to ICLR 2026_

### Official Review · Reviewer_FkkR · 2025-10-15

**Soundness:** 2
**Presentation:** 2
**Contribution:** 2
**Rating:** 2
**Confidence:** 4

**Summary:**

This work introduces ESNv2, revisits ESN with a diagonal complex-valued recurrence to enable parallelism, but offers limited real innovation beyond reinterpreting existing state-space and recurrent ideas.

The main contribution includes: 1. using a diagonal complex-valued linear recurrence that allows parallel computation. 2. adding a nonlinear mixing layer for expressivity while keeping the readout as the only trainable part.

**Strengths:**

1. This work has a clear motivation. The author figure out the the lack of parallelism in RCs and attempt to address it.

2. This work is well structured and easy to follow.

**Weaknesses:**

Although the method achieves faster training by eliminating sequential dependencies, this improvement comes at the expense of reduced model expressiveness, since the diagonal recurrence structure limits the network’s ability to capture complex temporal interactions.


The main problem is that the experiments only demonstrated mainly on simplistic synthetic benchmarks such as memory and forecasting tasks; moreover, comparisons with modern deep models like Transformers or LRUs appear shallow and unconvincing, as ESNv2 lacks the flexibility and scalability required for real-world applications.

Since the author has been benchmarking against the concept of SSM, I suggest the author test the real capability on text datasets.

Therefore, I believe this work does not meet the acceptance criteria

**Questions:**

1.  Please use more solid and credible experiments to demonstrate the effectiveness of the proposed solution. The currently used dataset is too old and too small. It's even a benchmark Lorenz96 published in 1996. While the largest classification dataset is MINST. This cannot represent the latest research progress at all.

2. These mini datasets cannot demonstrate the efficient and parallelized effects that the author claims to have proposed. In the current situation, I suggest the author consider ImageNet[1] as a baseline for classification task and PILE[2] for sequence modelling.


[1] Deng, Jia, et al. "Imagenet: A large-scale hierarchical image database." 2009 IEEE conference on computer vision and pattern recognition. Ieee, 2009.

[2] Gao, Leo, et al. "The pile: An 800gb dataset of diverse text for language modeling." arXiv preprint arXiv:2101.00027 (2020).

---

> ### Author Response · Authors · 2025-11-20
> **Author Rebuttal - Response to Reviewer FkkR**
>
> 1/2
>
> We thank the Reviewer for the time dedicated to reviewing our work. In particular, we thank the Reviewer for highlighting that our paper has a  “clear motivation” and that the manuscript is “well-structured and easy to follow”. We address each point below.
>
> Note that, following Reviewer prv8's feedback, **we renamed our model to ParalESN**. The manuscript, comments, and tables below use the new name accordingly.
>
> ---
>
> ### **Regarding the choice of benchmarks and datasets**
>
> We appreciate the Reviewer’s suggestions that datasets such as PILE or Imagenet would allow us to further showcase the computational efficiency of our approach. However, we think that, due to our model’s features,  its efficiency and parallelization capabilities are highlighted more by considering datasets with long sequences. Indeed, the parallelization that makes our approach computationally attractive occurs along the time dimension of the input data. Therefore, we are convinced that the experiments present in our work already adequately highlight this aspect. In particular, Fig. 2 (page 5) shows a comparison between time required to perform the recurrence between our approach and traditional RC, for sequence length up to $65k$. We report again the results below for completeness. Note how, at length $\approx 65k$, both ParalESN and ParalESN (deep) complete the recurrence in **three orders of magnitude less time** with respect to ESN and ESN (deep), respectively. Indeed, our approach scales (almost) constantly w.r.t. the sequence length rather than linearly. This is the contribution in terms of scaling and efficiency we bring to Reservoir Computing.
>
>
> | Sequence Length | 256 | 1024 | 4096 | 16384 | 65536 |
> |-----------------|-----|------|------|-------|-------|
> | ESN | 0.037 ± 0.001 | 0.146 ± 0.002 | 0.569 ± 0.004 | 2.619 ± 0.382 | 9.452 ± 0.522 |
> | ESN (deep) | 0.184 ± 0.020 | 0.700 ± 0.006 | 2.822 ± 0.047 | 11.794 ± 0.634 | 45.667 ± 3.129 |
> | **---** | **---** | **---** | **---** | **---** | **---** |
> | ParalESN | **0.003 ± 0.000** | **0.004 ± 0.000** | **0.005 ± 0.001** | **0.005 ± 0.001** | **0.007 ± 0.002** |
> | ParalESN (deep) | 0.015 ± 0.000 | 0.018 ± 0.001 | 0.022 ± 0.001 | 0.025 ± 0.001 | 0.042 ± 0.001 |
>
> Furthermore,Fig. 4 (page 8) shows a comparison of training time on all considered memory-based and forecasting tasks. Our approach completes training orders of magnitudes faster. Finally Fig. 3 (left, page 7) and Fig. 5 (page 10) highlights the trade-off between performance and efficiency.  Our approach outperforms traditional RC both in terms of performance and efficiency while being competitive with fully-trainable models. We argue that the shown experiments are sufficient to showcase the computational improvement of the approach. Moreover, our work is primarily aimed at improving the state of the art within Reservoir Computing. In this light, large scale language modeling or computer vision are beyond the scope of the paper. Instead, our focus is more on efficient and parallelizable dynamical systems modeling, which is central to RC research.
> However, we agree with the reviewer’s point of view that some of the initial benchmarks can appear too small-scale and limited. We included these datasets as they are standard initial benchmarks in the RC literature, enabling direct and fair comparison with classical ESNs.
>
> However, we agree that demonstrating applicability of our model on real-world datasets would further strengthen the work.
> For this reason, we included results on **modern, real-world forecasting datasets such as ETTh1, ETTh2, ETTm1,  ETTm2** [R1]. Below, we include a summary of the results on these new datasets. For more details refer to Table 2 (page 8) and Fig. 4 (page 8), for test set results and training time, respectively.
>
> | ↓ Test error | ETTh1 (×10⁻¹) | ETTh2 (×10⁻¹) | ETTm1 (×10⁻¹) | ETTm2 (×10⁻¹) |
> |-------------|-----------------|-----------------|-----------------|-----------------|
> | ESN | 9.1 ± 0.2 | 13.0 ± 1.7 | 6.7 ± 0.1 | 9.9 ± 7.3 |
> | ESN (deep) | 8.9 ± 0.1 | 9.6 ± 0.5 | 6.6 ± 0.0 | 6.0 ± 0.6 |
> | **---**          | **---**    | **---**      | **---**          | **---**        |
> | ParalESN | 8.9 ± 0.1 | 13.3 ± 1.5 | **6.5 ± 0.0** | **5.0 ± 0.1** |
> | ParalESN (deep) | **8.7 ± 0.1** | **9.2 ± 0.5** | **6.5 ± 0.0** | 5.3 ± 0.2 |
>
> | ↓ Training time (s) | ETTh1 | ETTh2 | ETTm1 | ETTm2 |
> |-------------|-------|-------|-------|-------|
> | ESN | 3.83 | 3.79 | 14.26 | 14.80 |
> | ESN (deep) | 15.68 | 16.03 | 27.07 | 38.89 |
> | **---**          | **---**    | **---**      | **---**          | **---**        |
> | ParalESN | **0.16** | **0.17** | **0.55** | **0.54** |
> | ParalESN (deep) | 0.21 | 0.21 | 0.62 | 0.58 |
>
> We believe that incorporating the ETT datasets meaningfully enhances the forecasting benchmark and offers a more realistic assessment of ParalESN ’s forecasting performance. We appreciate the reviewer’s observation, which prompted us to include these additional experiments.

---

> > ### Comment · Reviewer_FkkR · 2025-11-27
> >
> > I appreciate the author's response and I acknowledge that the article has some innovation.
> >
> > However, the additional data provided by the author is still difficult to convince me about my concerns regarding the usability of this article. The experiments under such a lightweight scale make it difficult to address my concerns. The model size is around 160k and there's no discussion about its scalability. Does the random layer still work with a sufficiently large size? And it seems more reasonable to replace the random parameter layer here with a learnable linear layer with dropout. At current scale, discussing effectiveness lacks sufficient motivation.
> >
> > I am willing to improve my score to 4 , but I still find it difficult to raise the score to above acceptance.

---

> ### Author Response · Authors · 2025-11-20
> **Author Rebuttal - Response to Reviewer FkkR**
>
> 2/2
>
> ### **Regarding the expressivity of our approach**
>
> On the expressivity of our model: as pointed out in Proposition 1 and Corollary 1, diagonal linear recurrence can be as powerful as classic RNN recurrence for our means, i.e. approximating a large class of functions defined on sequences of vectors. Moreover, thanks to the reparametrization that diagonalizes the transition matrix, the complex diagonal recurrence is effectively **mathematically equivalent** to a linear ESN with full recurrence matrix (Corollary 1).  The proofs in our work leverage previously established results on Reservoir Computing universality [R2]. In SSM literature, diagonal recurrence was proven to be as effective as full / structured recurrence, both theoretically and with thorough experiments [R3]. For these reasons, we clarify that our architectural simplifications do not reduce model expressiveness within the class of systems we aim to model.
>
> ---
>
> ###  **Final remarks**
> In light of the points expressed above, we hope that the reviewer’s will consider raising his score. Our work is positioned in the reservoir computing research area, and our choices of benchmarks and baselines reflect this focus. Moreover, we extended our empirical validation by including real-world modern forecasting datasets (ETT), which confirm practical relevance of our approach. Finally, our model is solidly grounded in ESN universality theory, together with modern diagonal SSMs theory, which demonstrate that diagonal recurrence is as expressive as full or structured recurrence. This directly contrasts with the reviewer’s concerns on loss of expressivity. Taken together, we believe these clarifications and additional results demonstrate that ParalESN provides a substantial and well-supported contribution to modern Reservoir Computing. We hope the reviewer will view the work in this light.
>
> ---
>
> ### **References**
>
> [R1] Haoyi Zhou, Shanghang Zhang, Jieqi Peng, Shuai Zhang, Jianxin Li, Hui Xiong, and Wancai Zhang. Informer: Beyond efficient transformer for long sequence time-series forecasting. In The Thirty Fifth AAAI Conference on Artificial Intelligence, AAAI 2021, Virtual Conference, volume 35, pages 11106–11115. AAAI Press, 2021.
>
> [R2] Grigoryeva, Lyudmila and Juan-Pablo Ortega. “Universal discrete-time reservoir computers with stochastic inputs and linear readouts using non-homogeneous state-affine systems.” J. Mach. Learn. Res. 19 (2017): 24:1-24:40.
>
> [R3] Ankit Gupta, Albert Gu, and Jonathan Berant. "Diagonal state spaces are as effective as structured state spaces." Advances in neural information processing systems 35 (2022): 22982-22994.

---

> ### Author Response · Authors · 2025-12-02
> **Author Rebuttal - Response to Reviewer FkkR**
>
> Thank you for engaging in the discussion and for recognizing that our paper introduces innovation. We acknowledge your concerns about the relatively lightweight scale of the experiments and would like to provide clarifications.
>
> We respond to each point in more detail below.
>
> ---
>
> ### **Regarding the lightweight scale of the experiments**
>
> We appreciate the opportunity to clarify the scope of our experiments. Our work primarily addresses the lack of parallelism in reservoir computing (RC). In other words, it targets computational scalability with respect to sequence length, rather than model size. For these reasons, we chose complex experiments on time series with long sequences to highlight the novelty and strength of our approach. The chosen benchmarks are commonly used to evaluate RC approaches, which are primarily applied to time series forecasting and classification.
>
> That said, to address your concerns about scalability, **we ran experiments on ParalESN up to $\approx 1M$ trainable parameters**. In the table below, we summarize these results. Notably, **ParalESN scales considerably better than traditional ESNs**, outperforming an ESN of $\approx 100K$ parameters with as few as $\approx 1K$ parameters. Additionally, ESNs run out of memory and are unable to train for the $1M$ trainable parameters setting. Indeed, traditional ESNs employ a dense $N_{h} \times N_{h}$ recurrent weight matrix, while ParalESN employs a diagonal $N_{h}$ parameterization. We are convinced that this experiment highlights the scalability of our approach to large sizes, especially when compared to traditional ESN.
>
> **↑ Accuracy (%) - (Reservoir size / Trainable parameters)**
>
> | psMNIST | ≈140 / ≈1K | ≈1.4K / ≈10K | ≈14K / ≈100K | ≈143K / ≈1M |
> |:---|:---:|:---:|:---:|:---:|
> | ESN | 61.6 ± 0.2 | 79.8 ± 0.1 | 81.2 ± 0.3 | Out of Memory |
> | **---** | **---** | **---** | **---** | **---** |
> | ParalESN | **87.2 ± 0.2** | **94.7 ± 0.1** | **95.9 ± 0.1** | **96.42 ± 0.1** |
>
>
> ---
>
>
> ### **Replacing the random layer with a learnable linear layer with dropout**
>
> Replacing the random (recurrent) layer with a learnable one would result in an improper use of RC. Indeed, the defining characteristic of RC approaches is that the recurrent layer is fixed and left untrained, and training is limited to a simple readout. The main reason for this is to achieve fast training and computational efficiency, and to bypass training via back-propagation through time (BPTT), which is slow and may bring issues related to vanishing/exploding (V/E) gradients. Therefore, replacing the random layer with a learnable one would result in the proposed approach losing the computational advantages that make it attractive, resulting in the same complexity as a fully-trainable RNN.
>
> That said, to address your concerns about the effectiveness of the random layer, **we ran experiments on replacing the random layer with a learnable layer with dropout**. In the table below, we summarize these results. We consider a learnable linear layer with dropout and, to expand the analysis, vanilla RNNs with smart initialization schemes (identity RNN-id [R1] and orthogonal RNN-ortho [R1]). The weights are trained via gradient descent. Notably, **ParalESN is the top-performing model despite leveraging a random layer**. Indeed, these fully-trainable models' performance may be hampered by issues related to optimization and/or V/E gradients.
>
> | psMNIST | Params. | ↑ Accuracy (%) |
> | --- | --- | --- |
> | RNN-id [R1] | ≈160K | 86.1 |
> | RNN-ortho [R1] | ≈160K | 89.3 |
> | learnable linear layer with dropout | ≈160K | 82.7 |
> | **---** | **---** | **---** | **---** |
> | ParalESN | ≈160K | **96.9** |
>
>
> ---
>
> ### **References**
>
> [R1] Voelker, Aaron, Ivana Kajić, and Chris Eliasmith. "Legendre memory units: Continuous-time representation in recurrent neural networks." Advances in neural information processing systems 32 (2019).

---

### Official Review · Reviewer_bxxD · 2025-10-17

**Soundness:** 2
**Presentation:** 3
**Contribution:** 2
**Rating:** 4
**Confidence:** 3

**Summary:**

The authors propose a new type of RC architecture that can have multiple layers and can process sequence data in parallel. They show the new architecture, ESNv2, can achieve comparable performance to ESN on many tasks while being computationally more efficient.

**Strengths:**

The idea of combining classical dynamical systems-inspired learning and contemporary large-scale modeling to bring reservoir computing into the deep learning regime is intriguing. If successful, it has the potential to scale up reservoir computing to tackle tasks that were previously out of reach. The problem is well motivated and the paper is well written.

**Weaknesses:**

* ESN and RC are most commonly used in time-series forecasting tasks. On this crucial task, ESNv2 does not improve on existing architectures in terms of performance.
* The benchmark on forecasting focuses on three very simple systems (Lorenz96, Mackey-Glass, NARMA), which may not be representative enough to draw robust conclusions about the forecasting capability of ESNv2 in general.
* The theoretical results seem to be direct applications of existing results.
* Things like the fading memory property are not defined.

**Questions:**

* Why is ESNv2 better than ESN at certain tasks (e.g., 1-D pixel-level classification) but not other tasks (e.g., forecasting chaotic systems)?
* One fundamental limitation of the traditional ESN is that the size of the reservoir cannot be scaled to billions of parameters due to the poor scaling of matrix inversion used in the training process (e.g., Ridge regression). Does ESNv2 address this limitation in any way? Is it possible to have a much larger reservoir in ESNv2 than in ESN?
* Figure 5 seems to be partially incompatible with Figure 3. Why do ESN and ESN (deep) have high efficiency in Figure 5 but low efficiency in Figure 3?
* It was mentioned that "Observe that even ESNv2 (deep), despite consisting of multiple reservoir layers, trains faster than a traditional, shallow ESN consisting of just one layer." My understanding is that both ESNv2 and ESN train through Ridge regression. So how does ESNv2 train faster than ESN with the same number of trainable parameters (i.e., when the dimension of the regression problem is the same)?

---

> ### Author Response · Authors · 2025-11-20
> **Author Rebuttal - Response to Reviewer bxxD**
>
> 1/2
>
> We thank the Reviewer for the positive feedback on our work, acknowledging that this line of work “has the potential to scale up reservoir computing to tackle tasks that were previously out of reach” and that “ the problem is well motivated and the paper is well written”, as well for the constructive criticisms to our work. In the following, we address the Reviewer’s concerns and their questions. We hope that, if satisfied with our clarifications, you will consider assigning a higher score.
>
> Note that, following Reviewer prv8's feedback, **we renamed our model to ParalESN**. The manuscript, comments, and tables below use the new name accordingly.
>
> ---
>
> ### **W1 and W2: Forecasting benchmarks**
> Thank you for highlighting the importance of time-series forecasting tasks when evaluating the performance of ESNs and RC approaches. We consider the results on time-series forecasting a crucial benchmark to showcase the competitiveness of our approach with respect to traditional shallow and deep RC.
>
> First, we would like to point out that the proposed approach's strength does not lie in consistently outperforming traditional RC in terms of performance. The goal is to propose a parallel and more efficient alternative to traditional RC, addressing one of its main limitations: having to process data sequentially. On top of this, ParalESN outperforms traditional RC in some memory-based and forecasting tasks, and in all time series and 1-D pixel-level classification tasks. More specifically, while ParalESN does not improve performance on some forecasting tasks when compared to traditional RC, it achieves comparable results while being significantly faster and more efficient. This advantage is highlighted in both Fig. 3 (left, page 7) and Fig. 4 (page 8). In the former, you may observe how ParalESN and ParalESN (deep) are considerably more efficient and performant with respect to their traditional counterparts when we consider the the overall efficiency and performance across all tasks. In the latter, you may observe that ParalESN is characterized by considerably lower training time, often by orders of magnitude.
>
> In order to address your concern, **we expanded the benchmark on time-series forecasting by adding a selection of real-world datasets from Electricity Transformer Temperature (ETT) [R1]: ETTh1, ETTh2, ETTm1, and ETTm2**. These are popular, recent and complex multi-variate forecasting benchmarks. In these tasks, **our approach consistently outperforms traditional RC while retaining the computational advantages that makes it attractive** (i.e., order of magnitudes lower training time). We report these new results on ETT datasets in the tables below. For more details refer to Table 2 (page 8) and Fig. 4 (page 8), for test set results and training time, respectively. We believe that the addition of these datasets from ETT provide a more representative evaluation of the forecasting capability of ParalESN.
>
> | ↓ Test error | ETTh1 (×10⁻¹) | ETTh2 (×10⁻¹) | ETTm1 (×10⁻¹) | ETTm2 (×10⁻¹) |
> |-------------|-----------------|-----------------|-----------------|-----------------|
> | ESN | 9.1 ± 0.2 | 13.0 ± 1.7 | 6.7 ± 0.1 | 9.9 ± 7.3 |
> | ESN (deep) | 8.9 ± 0.1 | 9.6 ± 0.5 | 6.6 ± 0.0 | 6.0 ± 0.6 |
> | **---**          | **---**    | **---**      | **---**          | **---**        |
> | ParalESN | 8.9 ± 0.1 | 13.3 ± 1.5 | **6.5 ± 0.0** | **5.0 ± 0.1** |
> | ParalESN (deep) | **8.7 ± 0.1** | **9.2 ± 0.5** | **6.5 ± 0.0** | 5.3 ± 0.2 |
>
> | ↓ Training time (s) | ETTh1 | ETTh2 | ETTm1 | ETTm2 |
> |-------------|-------|-------|-------|-------|
> | ESN | 3.83 | 3.79 | 14.26 | 14.80 |
> | ESN (deep) | 15.68 | 16.03 | 27.07 | 38.89 |
> | **---**          | **---**    | **---**      | **---**          | **---**        |
> | ParalESN | **0.16** | **0.17** | **0.55** | **0.54** |
> | ParalESN (deep) | 0.21 | 0.21 | 0.62 | 0.58 |
>
> ---
>
> ### **W3: Theoretical results**
> In this section, our goal was to provide precise guarantees of ParalESN expressivity by grounding the model in established theoretical results. To this end, **we derived a characterization of the Echo State Property (ESP) for the newly introduced architecture (Th. 1)** and connected existing universality results to our setting. Therefore, our contributions build upon the classical theoretical framework of ESNs, which we consider essential for establishing a solid foundation for ParalESN and clarifying why a diagonal, linear recurrence can match the expressivity of fully nonlinear recurrence in sufficiently rich function classes.

---

> ### Author Response · Authors · 2025-11-20
> **Author Rebuttal - Response to Reviewer bxxD**
>
> 2/2
>
> ---
> ### **W4: Definitions of fading memory filters**
> We thank the Reviewer for pointing out that a formal definition of fading memory and related concepts is missing. **We added Appendix B. Filters and Fading Memory Property** (pages 13-14) reviewing in more detail definitions of filters and related concepts.
>
> ---
>
> ### **Answers to Reviewer questions**
> - Q1: ParalESN exhibits near-optimal, long-term memory, as highlighted by the results achieved on the Memory Capacity (MemCap) task in Table 1 (page 8). We argue that, **to adequately solve time series and 1-D pixel-level classification tasks, it is crucial to possess long-term memory and recall all steps (or pixels) in the sequence**. On the other hand, in the considered forecasting tasks, having relatively short-term memory is often sufficient, depending on the considered delay. In such cases, even though ParalESN has a strong ability to recall past inputs, this advantage does not come into play because short-term memory is enough to effectively predict the next step in the sequence.
> - Q2: The issue related to the pseudo-inversion of the readout matrix is a fundamental limitation of training ridge regression in closed-form. Our approach does not address this, as these training procedures are a whole other problem that we chose not to tackle in this work. For very large reservoir sizes, one feasible approach is to train the linear/logistic regression in minibatches [R2]. Regarding the second part of the question, **ParalESN does allow for larger reservoirs when compared to ESN**. ParalESN ’s recurrent weight matrix is diagonal, resulting in a memory footprint of $N_{h}$, where $N_{h}$ is the reservoir size. On the other hand, traditional ESNs generally employ either fully-connected ($N_{h} \times N_{h}$) or sparse matrices. In the former scenario, the memory advantage of our approach is evident. In the latter scenario, while it may be possible via sparsity to reach a similar memory footprint of $N_{h}$, there is at least one main issue: limited expressivity. In our approach this is not an issue, as we preserve expressivity by having the diagonal weight matrix be in the complex space.
> - Q3: There are two factors that come into play here. In both figures, the **efficiency scores are normalized**. When ESN and ESN (deep) are compared against ParalESN and ParalESN (deep) in Fig. 3 (page 7), they are inefficient. When such a comparison includes fully-trainable models (e.g., Transformers, LSTMs) in Fig. 4 (page 8), these ESN models are still inefficient with respect to ParalESN and ParalESN (deep) but more efficient than these fully-trainable models. Thus, because the x-axis is a normalized efficiency score, they end up on the high-efficiency side simply because there are other models that are much, much more inefficient. Additionally, note that efficiency here (measured in terms of training time) heavily depends on the datasets considered and, specifically, on sequence length. In Fig. 3 (page 7), there are particularly long sequences, while in Fig. 4 (page 8) the results are computed solely on sMNIST and psMNIST, which have a relatively short sequence length of 784. As the sequence becomes shorter, the efficiency gap closes since the advantage of being able to parallelize along the time steps becomes less important. For reference, this is highlighted on a dedicated experiment in Fig. 2 (page 5).
> - Q4: Training ParalESN and traditional ESN involves two steps: (i) process data through the untrained reservoir and (ii) fit the readout layer. The training time corresponds to the time required to perform both of these steps. **While our method does not improve the efficiency of the "fitting" step**, which is the same in both approaches, **it does considerably improve the speed of the "processing" step**. Indeed, while traditional ESNs have to process data sequentially in the reservoir, ParalESN is able to process all time steps in parallel. For these reasons, ParalESN is able to train faster than ESN.
> ---
>
> ### **References**
>
> [R1] Haoyi Zhou, Shanghang Zhang, Jieqi Peng, Shuai Zhang, Jianxin Li, Hui Xiong, and Wancai Zhang. Informer: Beyond efficient transformer for long sequence time-series forecasting. In The Thirty Fifth AAAI Conference on Artificial Intelligence, AAAI 2021, Virtual Conference, volume 35, pages 11106–11115. AAAI Press, 2021.
>
> [R2] A. Dempster, F. Petitjean, and G. I. Webb. ROCKET: Exceptionally Fast and Accurate Time Series Classification Using Random Convolutional Kernels. In Data Mining and Knowledge Discovery, 34(5):1454–1495, 2020.

---

> ### Author Response · Authors · 2025-11-28
> **Official Comment by Authors**
>
> Dear Reviewer `bxxD`,
>
> First, we would like to reiterate our appreciation for the constructive feedback, and thank you for the time and effort you invested in reviewing our work.
>
> We are looking forward to your response. We hope that the additional experiments on time series forecasting, definitions on fading memory property, and improved rewriting have contributed positively to our work. We also hope that our comprehensive response and clarifications addressed your concerns. If so, we kindly ask the reviewer to consider adjusting the score accordingly.
>
> With sincere regards,
>
> The authors.

---

### Official Review · Reviewer_KJoa · 2025-10-29

**Soundness:** 3
**Presentation:** 3
**Contribution:** 3
**Rating:** 4
**Confidence:** 4

**Summary:**

The present manuscript proposes a structured method of Reservoir Computing, called ESNv2, that parallelize the input processing maintaining the Echo State Property, provided that the spectral radius of the diagonal weights is bounded, as proved in Theorem 1.
Moreover, for every ESN an equivalent ESNv2 in terms of expressivity can be found.
Evaluation on benchmarks is provided, comparing the proposed model with the traditional ESN and with popular recurrent models such as LSTM and Transformers.

**Strengths:**

The paper is well organized and easily readable.
The proposed model is supported by a sound, although simple, theoretical characterization (Theorem 1 and Proposition1).
The metrics evaluated in the experimental framework give a broad overview of the performance of the proposed model, also comparing with deep learning alternatives such as LSTMs and Transformers.

**Weaknesses:**

- Structured transforms in Reservoir Computing have already been proposed and investigated [1,2], but the present manuscript is totally missing a part of literature review in this regard, and therefore it lacks of a consequent comparison , in terms of performances and computational cost, with these structured methods.
- at line 99 it is claimed that the weight matrices "are generally sampled from a uniform distribution"; the claim lacks of bibliography references; moreover, in earlier works [3], it is suggested that the weights are initialized from Gaussian distributions;
- A reference and a consequent discussion to standard Deep Reservoir Computing is missing [4]
- The title, in my opinion, is disregarding the huge recent literature of works that are considering reservoir computing as a sound recurrent alternative to deep (backpropagation-trained) learning models for what concern hardware implementations [5].
Minor comments:
- In the proof of Theorem 1, there is a B in place of what it should be W_{in}; I believe it is so, because otherwise the proof wouldn't work.

[1] Dong, J., Ohana, R., Rafayelyan, M., & Krzakala, F. (2020). Reservoir computing meets recurrent kernels and structured transforms. Advances in Neural Information Processing Systems, 33, 16785-16796.
[2] D’Inverno, G. A., & Dong, J. (2025). Comparison of Reservoir Computing topologies using the Recurrent Kernel approach. Neurocomputing, 611, 128679.
[3] Verstraeten, D., Schrauwen, B., d’Haene, M., & Stroobandt, D. (2007). An experimental unification of reservoir computing methods. Neural networks, 20(3), 391-403.
[4] Gallicchio, C., Micheli, A., & Pedrelli, L. (2017). Deep reservoir computing: A critical experimental analysis. Neurocomputing, 268, 87-99.
[5] Gallicchio, C., & Soriano, M. C. (2025). Hardware friendly deep reservoir computing. Neural Networks, 108079.

**Questions:**

In view of what already said, my suggestion for the authors are listed as follows:
- the authors should integrate a substantial comparison with already existing structured transforms for RC in terms of formulation, theoretical guarantees and experimental validation;
- the authors may try to validate experimentally the performances in correspondence of Gaussian weight initialization;
- I would suggest either to revise the title, or to integrate the manuscript with a convincing argument to support the current title;
- all the missing references must be integrated in the manuscript.

---

> ### Author Response · Authors · 2025-11-20
> **Author Rebuttal - Response to Reviewer KJoa**
>
> We thank the Reviewer for the comments, questions and suggestions. We appreciate the positive comments highlighting that our paper is “well organized and easily readable”, that the “proposed model is supported by a sound theoretical characterization”, and that our experiments “give a broad overview of the performance of the proposed model, also comparing with deep learning alternatives”.
>
> Note that, following Reviewer prv8's feedback, **we renamed our model to ParalESN**. The manuscript, comments, and tables use the new name accordingly.
>
> We address each one of the Reviewer’s concerns below. We hope that, if satisfied with our clarifications, you will consider assigning a higher score.
>
> ---
> ### **W1: Previous works on structured transforms in Reservoir Computing**
> We thank you for the reference. **We added references on other structured reservoir approaches in the manuscript** [R1, R2]. While the referenced models share with ParalESN the general idea of structured, efficient transforms, the focal point of our approach is employing linear recurrence to have a parallelized training. In that sense, the diagonal expression of the transition function can be seen as a reparametrization that is useful in order to contain the computational complexity of the parallel scan, that would require matrix-matrix multiplications in the general case.
>
> ---
> ### **W2: Uniform vs Gaussian initialization**
> **We added a comparison between uniform and gaussian initialization** in ParalESN. See Appendix D.2, Fig. 6.(e) (page 17). We also **integrated the suggested reference** [R3]. Below, a summary of these results on the NARMA10 task. Uniform initialization achieves the lowest test error. Note that, for the Gaussian initialization the input scaling refers to the standard deviation, mean is fixed to 0.
>
> | Initialization | Input Scaling | ↓ Mean Test NRMSE |
> |----------------|---------------|-------------------|
> | Uniform | 0.01 | 0.072±0.007 |
> | Uniform | 0.1 | **0.039±0.011** |
> | Uniform | 1 | 0.041±0.008 |
> | Uniform | 10 | 0.042±0.010 |
> | **---** | **---** | **---** |
> | Gaussian | 0.01 | 0.044±0.013 |
> | Gaussian | 0.1 | 0.043±0.011 |
> | Gaussian | 1 | 0.045±0.008 |
> | Gaussian | 10 | 0.041±0.008 |
>
>
>
> ---
> ### **W3: Deep Reservoir Computing**
> We appreciate the time and effort in suggesting relevant missing references. We have **expanded our literature review section** dedicated to reservoir computing. Now, we adequately **reference previous work on Deep Echo State Networks (DeepESNs)** [R4].
>
> ---
>
> ### **W4: Regarding the manuscript's title**
> Thank you for this observation regarding the title. We understand the concern about not taking into account previous works that intersect reservoir computing and hardware implementations. On a similar note, taking into account feedback from other reviewers, we updated the model name to adequately highlight the core innovations of the proposed approach. That said, **we revisited the title accordingly: "ParalESN: Enabling parallel information processing in Reservoir Computing"**. Additionally, **we integrated the suggested reference in our manuscript** [R5].
>
> ---
>
> ### **W5: Typo**
> We apologize for the typos in the proof of Theorem 1. Thank you for bringing it to our attention. We have replaced $B$ with $W_{in}$ accordingly.
>
> ---
>
> ### **Answers to Reviewer questions**
> - Q1: **We have integrated the suggested works on structured transforms for RC in the Introduction section**. See W1.
> - Q2: **We added an empirical comparison between uniform and gaussian initialization** for ParalESN. See W2.
> - Q4: We acknowledge that the current manuscript's title does not take into account recent works at the intersection of reservoir computing and hardware implementations. **We have revised the title accordingly**. See W4.
> - Q4: **We integrated all references in the manuscript** [R1, R2, R3, R4, R5].
>
> ---
>
> ### **References**
>
> [R1] Dong, J., Ohana, R., Rafayelyan, M., & Krzakala, F. (2020). Reservoir computing meets recurrent kernels and structured transforms. Advances in Neural Information Processing Systems, 33, 16785-16796.
>
> [R2] D’Inverno, G. A., & Dong, J. (2025). Comparison of Reservoir Computing topologies using the Recurrent Kernel approach. Neurocomputing, 611, 128679.
>
> [R3] Verstraeten, D., Schrauwen, B., d’Haene, M., & Stroobandt, D. (2007). An experimental unification of reservoir computing methods. Neural networks, 20(3), 391-403.
>
> [R4] Gallicchio, C., Micheli, A., & Pedrelli, L. (2017). Deep reservoir computing: A critical experimental analysis. Neurocomputing, 268, 87-99.
>
> [R5] Gallicchio, C., & Soriano, M. C. (2025). Hardware friendly deep reservoir computing. Neural Networks, 108079.

---

> > ### Comment · Reviewer_KJoa · 2025-11-26
> > **Response to the authors**
> >
> > I would like to thank the authors for their careful revision of the manuscript.
> >
> > However, my question about the comparison with existing structured transforms in Reservoir Computing has been partially fulfilled, adding the suggested references but without comparing either theoretically nor experimentally with such structured transforms. I believe that this is a core comparison to be integrated in the manuscript in order to show a SOTA results, with respect to the important metrics (performance in terms of best error and, in this case, time complexity).
> >
> > Therefore, I cannot raise my score above the acceptance threshold.

---

> ### Author Response · Authors · 2025-11-27
> **Official Comment by Authors - Response to Reviewer KJoa**
>
> Thank you for engaging in the discussion, and providing additional detailed feedback. We acknowledge the importance of comparing our approach with existing structured transforms in Reservoir Computing (RC). **We added an experimental and theoretical comparison between ParalESN and other RC approaches based on structured transforms** in Appendix E (pages 17-19). We consider Structured Reservoir Computing (Structured RC) [R1] and Simple Cycle Reservoir (SCR) [R2]. Additionally, to highlight these core comparisons, **we updated the main body to summarize the findings**, see Section 3 (page 5).
>
> We respond to each point in more detail below. We hope that these clarifications address your concerns and merit a higher score.
>
> ---
>
> ### **Regarding a performance comparison in terms of best error.**
>
> Following your suggestion, **we included a test error comparison on time series forecasting benchmarks** in Appendix E.1 (page 18) and Table 8 (page 18). Below, a summary of these results on Lorenz, Mackey-Glass and NARMA tasks, for increasing look-ahead delay. Notably, **ParalESN consistently outperforms both SCR and Structured RC across all benchmarks**. In particular, compared to SCR and Structured RC, ParalESN achieves **lower test error  by an entire order of magnitude** on MG, MG84, and N10 and half the error on N30.
>
> | Forecasting | ×10⁻² Lz25 (↓) | ×10⁻² Lz50 (↓) | ×10⁻⁴ MG (↓) | ×10⁻² MG84 (↓) | ×10⁻² N10 (↓) | ×10⁻² N30 (↓) |
> |---|---|---|---|---|---|---|
> | ESN | **10.0±0.3** | 30.8±0.6 | **3.0±0.0** | **6.5±0.4** | **2.7±0.4** | **10.3±0.1** |
> | SCR | 22.1±0.4 | 35.6±0.8 | 18.7±4.2 | 32.2±3.1 | 11.1±0.2 | 16.1±0.9 |
> | Structured RC | 10.4±0.2 | 30.9±0.5 | 12.6±0.1 | 28.5±1.6 | 18.7±0.1 | 20.9±0.1 |
> | **---**          | **---**    | **---**      | **---**          | **---**    | **---**      | **---**      |
> | ParalESN | 10.3±0.4 | **28.8±0.4** | **3.0±0.0** | 7.8±0.4 | 3.7±1.0 | **10.3±0.1** |
>
>
> ---
>
>
> ### **Regarding a time complexity comparison.**
>
> Following your suggestion, **we included a time complexity analysis** in Appendix E.2 (pages 18-19) and Table 9 (page 19). Below, a summary of the computational complexity analysis. We report time complexity for computing (i) a single step and (ii) all steps in the recurrence. Notably, **ParalESN allows for a number of operations that scales linearly with the reservoir size** $N_{h}$, thanks to its diagonal transition matrix. Furthermore, thanks to its linear recurrence, **ParalESN's time complexity does not depend linearly on the sequence length but logarithmically**, as it is possible to compute all time steps in parallel via associative scan rather than sequentially.
>
> | Model | Single step | Whole sequence |
> |---|---|---|
> | ESN | $\mathcal{O}(N_h^2)$ | $\mathcal{O}(LN_h^2)$ |
> | SCR | $\mathcal{O}(N_hN_i)$ | $\mathcal{O}(LN_hN_i)$ |
> | Structured RC | $\mathcal{O}(N_h\log N_h)$ | $\mathcal{O}(LN_h\log N_h)$ |
> | **---**          | **---**    | **---**      |
> | ParalESN | $\mathcal{O}(N_hN_i)$ | $\mathcal{O}(\log{(L)}N_hN_i)$ |
>
> ---
>
> ### **References**
> [R1] Dong, Jonathan, et al. "Reservoir computing meets recurrent kernels and structured transforms." Advances in Neural Information Processing Systems 33 (2020): 16785-16796.
>
> [R2] Rodan, Ali, and Peter Tino. "Minimum complexity echo state network." IEEE transactions on neural networks 22.1 (2010): 131-144.

---

> > ### Comment · Reviewer_KJoa · 2025-11-28
> > **Response to the authors**
> >
> > I would like to thank again the authors for bringing on additional experiments and analysis on the comparison with existing structured RC models. The complexity analysis seems correct and the experiments suggest the superior performance of ParalelESN, therefore I will now raise my score to 8.

---

### Official Review · Reviewer_prv8 · 2025-11-01

**Soundness:** 3
**Presentation:** 3
**Contribution:** 2
**Rating:** 4
**Confidence:** 4

**Summary:**

The paper presents a framework, ESNv2, which introduces the diagonal linear recurrence in the complex space into traditional RC systems. The paper also provides a theoretical analysis and empirical validation, demonstrating the model's efficiency and competitive performance compared to both classical RC and some deep learning models (LSTM, Transformer, LRU) on sequential MNIST tasks.

**Strengths:**

1. The introduction of diagonal linear recurrence in the complex space is a contribution to RC that allows for parallelization during training.
2. The paper includes a solid theoretical foundation, proving that ESNv2 preserves the Echo State Property (ESP) and universality guarantees, which strengthens its credibility.
3. The paper is well-structured, with clear explanations and logical flow from theoretical concepts to practical experimentation.

**Weaknesses:**

1. The name "ESNv2" does not adequately reflect the core innovations of the model. A more descriptive name that captures the essence of the diagonal linear recurrence and its parallelization capabilities would enhance clarity and impact.
2. While the paper compares ESNv2 with various models, a more detailed comparative analysis with a wider range of state-of-the-art deep learning models, particularly in terms of specific applications, could enhance the discussion.
3. The work does not deeply survey and discuss related works on ESN variants and Deep ESN, which could provide valuable context and highlight the novelty of ESNv2.
4. The absence of a comparison with Mamba and related state space models is a notable gap. Including this comparison would enrich the findings and establish ESNv2's position more clearly within the landscape of contemporary models.
5. The paper does not discuss how key parameters from traditional ESNs, such as input scaling and spectral radius, are set and their influence on the new model (missing in the methodology section). A detailed examination of these parameters would improve understanding of ESNv2's behavior and performance.
6. The paper lacks a comparison of ESNv2's performance on real-world time series prediction tasks. Adding such analysis would provide practical insights into the model's applicability and effectiveness in real-world scenarios.
7. The method of hyperparameter tuning is mentioned, but more details on the specific impact of different hyperparameters on performance could be beneficial for reproducibility and practical implementation.

**Questions:**

In the part of Weaknesses.

---

> ### Author Response · Authors · 2025-11-20
> **Author Rebuttal - Response to Reviewer prv8**
>
> 1/3
>
> We appreciate the Reviewer's careful reading of our manuscript and their constructive feedback. In particular, we thank the Reviewer for highlighting that our paper "introduces the diagonal linear recurrence in the complex space into traditional RC systems", that this “is a contribution to RC that allows for parallelization during training”, that the paper “includes a solid theoretical foundation, proving that our approach preserves the Echo State Property (ESP) and universality guarantees, which strengthens its credibility”, and that the paper is “well-structured, with clear explanations and logical flow from theoretical concepts to practical experimentation”.
>
> We address each of the Reviewer's questions below. We hope that these clarifications address your concerns and merit a higher score.
>
> ---
>
> ### **W1: Model name**
> We appreciate the Reviewer's thoughtful concern regarding the chosen model name, "ESNv2". We now recognize that such a name does not adequately reflect the core innovations of the proposed model. To address this, **we renamed "ESNv2" to "Parallel ESN (ParalESN)"**.
>
> An important note is that, while the diagonal linear recurrence is itself a core innovation, its end goal is to enable parallelization during training, since we are able to compute all time steps in the recurrence in parallel via associative scan. Therefore, we decided to prioritize highlighting the parallelization aspect of the proposed model in its new name, rather than the diagonal linear recurrence.
>
> ---
>
> ### **W2: Comparison with a wider range of state-of-the-art deep learning models.**
> We appreciate the suggestion to expand our comparative analysis. Inspired by the Reviewer's comment, **we performed new experiments on sequential MNIST (sMNIST) and permuted sequential MNIST (psMNIST) using Mamba**. We report the updated results for these datasets in the tables below. For more details, see the updated Table 4 (page 9) and the updated Fig. 5 (page 10). **Mamba slightly outperforms ParalESN/ParalESN (deep) on sMNIST** by $+1.44\%$ test accuracy, but **it considerably underperforms on psMNIST** by approx. $-5\%$ test accuracy. On top of this, **our models are considerably more efficient, with less training time, emissions, and energy consumption by one order of magnitude**.
>
> | sMNIST           | ↑ Accuracy | ↓ Time (min) | ↓ Emissions (kg) | ↓ Energy (kWh) |
> |------------------|------------|--------------|------------------|----------------|
> | LSTM             | 98.53    | 86.54        | 0.53             | 1.61           |
> | Transformer      | 98.44      | 154.78       | 1.00             | 3.02           |
> | LRU              | **99.00**  | 29.07        | 0.18             | 0.57           |
> | Mamba            | 98.50      | 29.01        | 0.19             | 0.57           |
> | ESN              | 83.65      | 4.36         | **0.02**         | 0.07           |
> | ESN (deep)       | 89.67      | 8.30         | 0.05           | 0.14           |
> | **---**          | **---**    | **---**      | **---**          | **---**        |
> | ParalESN            | 97.06      | **2.91**     | **0.02**         | **0.05**       |
> | ParalESN (deep)     | 94.84      | 3.35       | **0.02**         | 0.06         |
>
> | psMNIST          | ↑ Accuracy | ↓ Time (min) | ↓ Emissions (kg) | ↓ Energy (kWh) |
> |------------------|------------|--------------|------------------|----------------|
> | LSTM             | 92.37      | 83.37        | 0.52             | 1.57           |
> | Transformer      | 97.45      | 155.18       | 1.00             | 3.02           |
> | LRU              | **97.80**  | 33.80        | 0.21             | 0.63           |
> | Mamba            | 92.70      | 37.76        | 0.24             | 0.73           |
> | ESN              | 79.75      | 4.40         | 0.02           | 0.07           |
> | ESN (deep)       | 82.24      | 7.41         | 0.04             | 0.12           |
> | **---**          | **---**    | **---**      | **---**          | **---**        |
> | ParalESN            | 96.88      | **2.49**     | **0.01**         | **0.04**       |
> | ParalESN (deep)     | 97.65    | 2.90       | 0.02          | 0.05         |
>
>
> ---
> ### **W3: Discussion on ESN variants and DeepESN.**
>
> Thank you for raising this important point. Following your suggestions, **we expanded our literature review to include include DeepESN [R1] and Residual Echo State Networks (ResESN) [R2]** - a recent ESN variation based on orthogonal residual connections.
>
> As a comment, both DeepESNs and ResESNs, despite introducing architectural innovations compared to traditional ESNs, rely on a non-linear recurrence which is inherently non parallelizable. Therefore, these systems must process data sequentially as they are not able to parallelize computations along the time dimension. Our work introduces linear diagonal recurrence to RC to address this limitation.

---

> ### Author Response · Authors · 2025-11-20
> **Author Rebuttal - Response to Reviewer prv8**
>
> 2/3
>
> ### **W4: Comparison with Mamba and related state space models.**
>
> Please refer to point **W2**.
>
> ---
>
> ### **W5: Influence of hyperparameters and how they are set.**
> We do discuss how key hyperparameters from traditional RC, such as the input scaling or the spectral radius, are set and used. Both **in the Methodology (page 6) and in Appendix D (page 16), we mention running model selection via Bayesian search and selecting the best set of hyperparameters leveraging a validation set**. Unfortunately, there is no magic here, and model selection is crucial in determining to which values ParalESN ’s hyperparameters are set for the task at hand. Additionally, **in Section 3 (page 4) we discuss how the weight matrices and bias vector are initialized based on scalings and spectral radius**. We sample their entries randomly from a uniform distribution, as in traditional RC. For the recurrent weight matrix, we sample the diagonal elements by “drawing radii uniformly from $[\rho_{min}, \rho_{max}]$ and phases uniformly from $[phase_{min}, phase_{max}]$, then converting to complex form and incorporating the leaky rate to shift the eigenvalues center”.
>
> That said, to address your concerns about the lack of discussion on ParalESN’s hyperparameters impact on performance, **we added experiments exploring the impact of input scaling, leaky rate, spectral radius, and phase** on the NARMA10 task. Below, we include a table summarizing these results for each hyperparameter. For more details, refer to the violin plots in Appendix D.2, Fig. 6 (page 17). Notably, uniform initialization for the input weights matrix leads to lower test error compared to Gaussian initialization. The leaky rate rate is particularly important for optimal performance, with leakage $0.1$ and $0.5$ leading to one order of magnitude higher test error when compared to values such as $0.9$ and $1.0$.
>
>
> | Initialization | Input Scaling | ↓ Mean Test NRMSE |
> |----------------|---------------|-------------------|
> | Uniform | 0.01 | 0.072±0.007 |
> | Uniform | 0.1 | **0.039±0.011** |
> | Uniform | 1 | 0.041±0.008 |
> | Uniform | 10 | 0.042±0.010 |
> | **---** | **---** | **---** |
> | Gaussian | 0.01 | 0.044±0.013 |
> | Gaussian | 0.1 | 0.043±0.011 |
> | Gaussian | 1 | 0.045±0.008 |
> | Gaussian | 10 | 0.041±0.008 |
>
>
> | Leaky Rate | ↓ Mean Test NRMSE |
> |------------|-------------------|
> | 0.1 | 0.193±0.001 |
> | 0.5 | 0.128±0.001 |
> | 0.9 | **0.040±0.009** |
> | 1.0 | 0.051±0.020 |
>
>
>
> | Spectral radius (max) | ↓ Mean Test NRMSE |
> |----------------------|-------------------|
> | 0.1 | 0.179±0.001 |
> | 0.5 | **0.039±0.017** |
> | 0.9 | 0.045±0.012 |
> | 1.0 | 0.053±0.008 |
>
> | Spectral Radius (min) | ↓ Mean Test NRMSE |
> |----------------------|-------------------|
> | 0 | 0.048±0.012 |
> | 0.1 | 0.041±0.006 |
> | 0.5 | **0.047±0.018** |
> | 0.9 | 0.062±0.006 |
> | 1.0 | 0.116±0.004 |
>
> | Phase (max) | ↓ Mean Test NRMSE |
> |-------------|-------------------|
> | π/2 | 0.098±0.003 |
> | π | 0.044±0.010 |
> | 2π | **0.038±0.013** |
>
> ---
>
> ### **W6: Comparison on real-world time series predictions tasks.**
>
> We share the Reviewer's concern related to missing comparison on real-world time series prediction tasks. As a starting point, we intentionally focused on time series prediction tasks that are particularly popular when evaluating reservoir computing and ESNs (e.g., NARMA, Lorenz, Mackey-Glass, ...). We believe that these tasks were a fundamental step needed to understand whether the proposed approach was able to achieve comparable performance to traditional RC on top of its computational advantage. Additionally, we would like to point out that the time series classification benchmarks from UEA/UCR in Table 3 (page 9) are also real world tasks on time series. That said, following your suggestions, **we expanded our experiments by including benchmarks on real-world, time-series forecasting tasks**. Specifically, **we added the following tasks from Electricity Transformer Temperature (ETT) [R3]: ETTh1, ETTh2, ETTm1, and ETTm2**. These datasets are complex, real-world, multi-variate forecasting benchmarks on time series.
>
> In these tasks, our approach consistently outperforms traditional RC while retaining the computational advantages that makes it attractive (i.e., order of magnitudes lower training time). We report these new results on ETT datasets in the tables below. For more details refer to Table 2 (page 8) and Fig. 4 (page 8), for test set results and training time, respectively.
>
> | ↓ Test error | ETTh1 (×10⁻¹) | ETTh2 (×10⁻¹) | ETTm1 (×10⁻¹) | ETTm2 (×10⁻¹) |
> |-------------|-----------------|-----------------|-----------------|-----------------|
> | ESN | 9.1 ± 0.2 | 13.0 ± 1.7 | 6.7 ± 0.1 | 9.9 ± 7.3 |
> | ESN (deep) | 8.9 ± 0.1 | 9.6 ± 0.5 | 6.6 ± 0.0 | 6.0 ± 0.6 |
> | **---**          | **---**    | **---**      | **---**          | **---**        |
> | ParalESN | 8.9 ± 0.1 | 13.3 ± 1.5 | **6.5 ± 0.0** | **5.0 ± 0.1** |
> | ParalESN (deep) | **8.7 ± 0.1** | **9.2 ± 0.5** | **6.5 ± 0.0** | 5.3 ± 0.2 |

---

> ### Author Response · Authors · 2025-11-20
> **Author Rebuttal - Response to Reviewer prv8**
>
> 3/3 continued
>
> | ↓ Training time (s) | ETTh1 | ETTh2 | ETTm1 | ETTm2 |
> |-------------|-------|-------|-------|-------|
> | ESN | 3.83 | 3.79 | 14.26 | 14.80 |
> | ESN (deep) | 15.68 | 16.03 | 27.07 | 38.89 |
> | **---**          | **---**    | **---**      | **---**          | **---**        |
> | ParalESN | **0.16** | **0.17** | **0.55** | **0.54** |
> | ParalESN (deep) | 0.21 | 0.21 | 0.62 | 0.58 |
> ---
>
> ### **W7: Details on performance impact of different hyperparameters**
>
> Please refer to point **W5**.
>
> ---
>
> ### **References**
> [R1] Gallicchio, Claudio, Alessio Micheli, and Luca Pedrelli. "Deep reservoir computing: A critical experimental analysis." Neurocomputing 268 (2017): 87-99.
>
> [R2] Ceni, Andrea, and Claudio Gallicchio. "Residual Echo State Networks: Residual recurrent neural networks with stable dynamics and fast learning." Neurocomputing 597 (2024): 127966.
>
> [R3] Haoyi Zhou, Shanghang Zhang, Jieqi Peng, Shuai Zhang, Jianxin Li, Hui Xiong, and Wancai Zhang. Informer: Beyond efficient transformer for long sequence time-series forecasting. In The Thirty Fifth AAAI Conference on Artificial Intelligence, AAAI 2021, Virtual Conference, volume 35, pages 11106–11115. AAAI Press, 2021.

---

> ### Author Response · Authors · 2025-11-28
> **Official Comment by Authors**
>
> Dear Reviewer `prv8`,
>
> First, please allow us to express our gratitude for your review, and the time and effort you invested in it.
>
> We are looking forward to your response. We hope that our comprehensive responses and inclusion of additional details have contributed positively to our work. If so, we kindly ask the reviewer to consider adjusting the score accordingly.
>
> With sincere regards,
>
> The authors.

---

### Author Response · Authors · 2025-11-20
**Message to all Reviewers**

Thank you for the comments, questions, and suggestions. We particularly appreciate your recognition that our work addresses a key limitation of reservoir computing, the lack of parallelism, by introducing diagonal linear recurrence to enable parallel training. We also appreciate that you found our paper well-structured and with clear motivations, and that you noted the theoretical analysis strengthens our claims.

We have responded to individual reviewer comments.

To avoid confusion, **we point out that our model "ESNv2" has been renamed to "Parallel ESN (ParalESN)"**, following Reviewer prv8's feedback. **We will refer to our approach as ParalESN going forward**. The manuscript has been updated accordingly.

---

All in all, we believe that this work represents a novel contribution to reservoir computing and provides robust empirical results complemented by a sound theoretical analysis. We also believe that this work would benefit the ICLR community in this venue. We hope that the Reviewers agree, particularly given the new experiments, improved rewriting, and clarifications, and see fit to raise their scores.

---

### Comment · Area_Chair_RFno · 2025-11-27

Dear reviewers,

A reminder that the discussion phase will end in a few days (**December 2**). Engaging with the author's rebuttal is essential to address all potential concerns before our final discussion stage.

Thanks,
The AC

---

### Author Response · Authors · 2025-12-02
**Rebuttal Summary**

Dear AC and Reviewers,

We thank you all for your efforts. We are pleased that **all reviewers who engaged in the discussion decided to raise their scores**. To facilitate the final decision of the new Area Chair, we summarize the interactions with each reviewer below.

---

**Reviewer `KJoa` (Score: 4 → 8).** They appreciated the paper's readability, theoretical characterization, and experimental framework. They suggested additional experiments and time complexity analyses on structured RC, integrating references, and revising the title. As requested:
- We added new experiments and time complexity analyses on structured RC.
- We integrated all suggested references in the manuscript.
- We revised the title.

Given the changes and clarifications, *the reviewer was satisfied and increased their score from 4 to 8*.

---

**Reviewer `FkkR` (Score: 2 → 4).** They appreciated the paper's clear motivation and the novelty of addressing the lack of parallelism in reservoir computing. They requested clarifications about the expressivity of the diagonal linear recurrence and additional experiments. As requested:
- We clarified that the expressivity of our approach is preserved, as pointed out in Proposition 1 and Corollary 1.
- We added additional experiments on complex real-world forecasting datasets and scalability.

Given the changes and clarifications, *the reviewer raised their score from 2 to 4*. We believe that, had the security breach not happened, we could have been able to clarify the reviewer's remaining concerns, possibly leading to an above-acceptance score.

---

**Reviewer `prv8` (Score: 4 → N/A, prevented from responding due to the security breach).** They appreciated the novelty of our approach in enabling parallel training in reservoir computing and the solid theoretical foundation. They suggested revising the model's name, including additional experiments, and integrating additional references. As requested:
- We revised the model's name.
- We added new experiments on Mamba, complex real-world forecasting benchmarks, and the effect of model’s hyperparameters.
- We integrated all suggested references in the manuscript.

We fully addressed all of the reviewer’s weaknesses/questions. Unfortunately, *the reviewer did not have the opportunity to engage in the discussion before the security breach*. Thus the score remained unchanged at 4. However, we believe that, given the new experiments and clarifications, a higher score was more than likely.

---

**Reviewer `bxxD` (Score: 4 → N/A, prevented from responding due to the security breach).** They recognized that our approach brings reservoir computing into the deep learning regime, and its potential for scaling up reservoir computing to tasks that were previously out of reach. They suggested additional experiments on time series forecasting, including definitions related to the fading memory property, and requested clarifications. As requested:
- We added experiments on complex real-world forecasting benchmarks.
- We added definitions of fading memory and related concepts.
- We responded to and clarified all of the reviewer's questions.

We fully addressed all of the reviewer’s weaknesses/questions. Unfortunately, *the reviewer did not have the opportunity to engage in the discussion before the security breach*. Thus the score remained unchanged at 4. However, we believe that, given the new experiments and clarifications, a higher score was more than likely.

---

### Meta-Review · Area_Chair_fDtS · 2026-01-01

**Summary:**

This paper introduces the Parallel Echo State Network (ParalESN). This framework enables the construction of efficient reservoirs with diagonal linear recurrence in the complex space, which can be parallelized during training. It presents both a theoretical analysis and empirical validation, demonstrating the model's efficiency and competitive performance relative to classical RC and deep learning models such as LSTM, Transformer, and LRU on sequential MNIST tasks.

**Reviewer Concerns:**

The reviewers, prv8, KJoa, FkkR, and bxxD, raise concerns that the paper needs a more descriptive name, broader comparisons, a deeper discussion of related work, an analysis of ESN parameters, real-world evaluations, and clearer hyperparameter details. Some of the concerns are addressed in the rebuttal, such as adding experiments on complex real-world forecasting benchmarks and defining fading memory. Some concerns about the usability of this method remain.

While the paper mentions deep ESN, it lacks a systematic review of the related literature. The time-series forecasting evaluation is limited, using only a few ETT datasets not included in the updated PDF. The new forecasting experiments still rely on ESN as the baseline, which is insufficient. The hyperparameter discussion is shallow, with no analysis of deep-layer properties. Since a diagonal linear recurrence in complex space is key, renaming the model "Parallel ESN" may be misleading.

Overall, although this paper has some merits, and the authors provided a thorough rebuttal, it still falls short of the acceptance standards for a top-tier conference.

**Reviewer Scores:**

The initial overall scores from reviewers prv8, KJoa, bxxD, and FkkR were 4, 4, 4, and 2, with corresponding confidence scores of 4, 4, 3, and 4. After reviewing the full rebuttal and the authors’ responses, KJoa and FkkR raised their scores to 8 and 4, respectively. I think the remaining two reviewers would keep their scores.

---

### Decision · Program_Chairs · 2026-01-26

Reject